# Structure of the native γ-tubulin ring complex capping spindle microtubules

Tom Dendooven [1,6] ✉, Stanislau Yatskevich [1,5,6] ✉, Alister Burt[1,5], Zhuo A. Chen [2], Dom Bellini [1], Juri Rappsilber [2,3,4], John V. Kilmartin [1] ✉ & David Barford [1] ✉

Microtubule (MT) filaments, composed of α/β-tubulin dimers, are fundamental to cellular architecture, function and organismal development. They are nucleated from MT organizing centers by the evolutionarily conserved γ-tubulin ring complex (γTuRC). However, the molecular mechanism of nucleation remains elusive. Here we used cryo-electron tomography to determine the structure of the native γTuRC capping the minus end of a MT in the context of enriched budding yeast spindles. In our structure, γTuRC presents a ring of γ-tubulin subunits to seed nucleation of exclusively 13-protofilament MTs, adopting an active closed conformation to function as a perfect geometric template for MT nucleation. Our cryo-electron tomography reconstruction revealed that a coiled-coil protein staples the first row of α/β-tubulin of the MT to alternating positions along the γ-tubulin ring of γTuRC. This positioning of α/β-tubulin onto γTuRC suggests a role for the coiled-coil protein in augmenting γTuRC-mediated MT nucleation. Based on our results, we describe a molecular model for budding yeast γTuRC activation and MT nucleation.

Microtubule (MT) filaments are essential cellular structures[1]. MT filaments assemble the mitotic spindle apparatus responsible for chromosome segregation, whereas the MT-based cytoskeletal network mediates dynein and kinesin-based intracellular transport. MTs are formed by the dynamic oligomerization and depolymerization of α/β-tubulin dimers in a head-to-tail fashion, with α-tubulin exposed at the 'minus' end of MTs and β-tubulin at the more dynamic MT 'plus' end[2]. In vitro, de novo α/β-tubulin polymerization to form MTs is kinetically limited and occurs only at high tubulin concentrations[3–5], whereas in cells the evolutionarily conserved γTuRC templates efficient MT nucleation at MT organizing centers (MTOCs) such as the vertebrate centrosome and yeast spindle pole body (SPB) that assemble mitotic spindles[6–9]. Previous structural studies of native *Xenopus laevis*[10] and *Homo sapiens*[11,12] γTuRCs, as well as recombinant *H. sapiens*[13–15] and *Saccharomyces cerevisiae*[16–18] γTuRCs showed that the complex forms a large conical structure. *S. cerevisiae* γTuRC is built from a helical repeat of seven 'V'-shaped copies of the tetrameric γ-tubulin small complex (γTuSC)[18,19], composed of Spc97/GCP2 (SPB component 97/γ-tubulin complex 2) and Spc98/GCP3 subunits each bound to a γ-tubulin molecule[16,17]. Spc110 (vertebrate CDK5RAP2 ortholog) links adjacent γTuSCs via its centrosomin motif 1 (CM1), and Spc110 is also required for γTuRC recruitment and assembly at the SPB[18,20–23]. γTuRC contains 14 γ-tubulin molecules, but it templates nucleation of 13-protofilament (PF) MTs. Most vertebrate γTuRCs exhibit open conformations and reveal a geometrical mismatch between the γ-tubulin ring and the MT lattice, and are poor templates for MT nucleation in vitro. Notably, the reported higher affinity of γ-tubulin for α/β-tubulin, compared to α/β-tubulin self-association, is neither sufficient to nucleate MTs nor for γTuRC to transition to the closed conformation[7]. Structures of yeast γTuSC filaments exhibit a spectrum of conformational states, and appear to sample 'open' and near 'closed' conformations in solution[18]. An engineered yeast γTuSC filament stabilized with disulfide cross-links

[1]MRC Laboratory of Molecular Biology, Cambridge, UK. [2]Technische Universität Berlin, Chair of Bioanalytics, Berlin, Germany. [3]Si-M/'Der Simulierte Mensch', Technische Universität Berlin and Charité, Universitätsmedizin Berlin, Berlin, Germany. [4]Wellcome Centre for Cell Biology, University of Edinburgh, Edinburgh, UK. [5]Present address: Genentech, South San Francisco, CA, USA. [6]These authors contributed equally: Tom Dendooven, Stanislau Yatskevich. ✉e-mail: tdendooven@mrc-lmb.cam.ac.uk; yatskevs@gene.com; jvk@mrc-lmb.cam.ac.uk; dbarford@mrc-lmb.cam.ac.uk

helped to lock a near 'closed' conformation of the γTuSC filament[17]. Disulfide cross-linking also enhanced MT nucleation efficiency of yeast γTuRC two- to threefold, yet the α/β-tubulin concentrations required for efficient nucleation were still above physiological levels[17]. This suggests that matching the geometry of γTuRC to the geometry of MT is an important but perhaps not sufficient step to control MT nucleation. Reconstitution and cell biology experiments also defined a critical role for MT polymerases in γTuRC-mediated MT nucleation[12,24–27]. These studies present an intriguing puzzle: how is efficient MT nucleation in cells achieved from a seemingly imperfect template? A number of mechanisms have been proposed to explain the high efficiency of MT nucleation at endogenous γTuRC: activation by effector molecules such as MT polymerases, tubulin lattice-mediated activation and phosphorylation-driven activation[7]. It is, however, unknown to what degree and how each of these mechanisms, if at all, contributes to γTuRC activation and MT nucleation.

In this article, to understand how γTuRC mediates nucleation of exclusively 13-PF MTs, we determined the structure of the active native γTuRC as part of enriched yeast mitotic spindles using cryo-electron tomography (cryo-ET). In our structure, γTuRC is observed as a perfect geometric template bound at the minus end of the complete MT lattice. We describe two additional novel cryo-ET densities in our reconstructions: (1) long coiled-coil-like proteins spanning Spc98, γ-tubulin and the first α/β-tubulin array of the MT lattice, where these coiled coils staple the first row of α/β-tubulin to the γ-tubulin ring, and (2) a luminal bridge density that probably stabilizes the oligomeric state of the γTuRC capping MT minus ends. Our structure of a native γTuRC capping a MT expands our understanding of the molecular mechanism of γTuRC-mediated MT nucleation.

## Results

### γTuRC forms a perfect helical template for MT nucleation

We prepared enriched endogenous *S. cerevisiae* (strain NCYC74) SPBs, the MTOCs in yeast, for cryo-ET studies (Fig. 1a,b, Extended Data Fig. 1a and Supplementary Video 1). The protein sequences of the γTuRC components of this strain are at least 95% identical to the canonical *S. cerevisiae* S288c. Canonical structures such as the outer, intermediate and central plaques[28,29], in addition to ~25 nuclear MTs (nMTs) per SPB, were readily observed in the SPB tomograms (Fig. 1b and Supplementary Video 1). This matches on average the number of MTs per spindle pole reported for intact *S. cerevisiae* cells[30], suggesting that the majority of MTs remain stably associated with the SPBs during enrichment. Additionally, the MT lattices are unperturbed (as discussed later during MT subtomogram averaging), and virtually all MTs are attached via their minus ends, to the inner plaque of the SPB (Fig. 1b).

We performed subtomogram averaging of the SPB-proximal MT minus ends, capped by γTuRC, and the MT lattice itself using a tailored processing pipeline (Extended Data Fig. 1b–d). We determined a consensus reconstruction of the native yeast γTuRC, capping the α/β-tubulin lattice at the minus end, to a global resolution of 9.2 Å (Fig. 1c,d, Table 1, Extended Data Fig. 1b,c and Supplementary Video 1). Subboxing of the γTuRC at γTuSC positions, followed by focused refinements, improved the resolution to 7–9 Å resolution (Extended Data Fig. 1b,c). In parallel, we also reconstructed the native yeast MT lattice at 6.6 Å resolution (Table 1 and Extended Data Fig. 1b,c). Using available high-resolution structures of yeast γTuSC[18,31] and MTs[32], we generated a complete molecular model of γTuRC capping mitotic spindle MT minus ends (Fig. 1c,d and Supplementary Video 1). In our tomograms and during subtomogram averaging, we did not observe γTuRCs devoid of MTs or incomplete γTuRC complexes, nor did we observe detached γTuRC molecules or uncapped MTs, suggesting that only complete γTuRC complexes attached to the SPB nucleate MTs in the native system. This is consistent with observations that recruitment of γTuRC to the SPBs and MT nucleation are closely coupled and regulated processes[16,20,22,23,33].

In our γTuRC reconstruction, seven γTuSC protomers assemble side-on to generate a pseudo-helical γTuRC at the base of the MT lattice, consisting of 14 spokes (formed by Spc98:γ-tubulin and Spc97:γ-tubulin of γTuSC promoters) (Figs. 1c,d and 2a, and Extended Data Fig. 2a). As spoke 14 is positioned directly above spoke 1, γTuRC presents 13 γ-tubulin molecules for MT nucleation (spokes 2–14) (Extended Data Fig. 2a,b). The stoichiometry of the native budding yeast γTuRC is identical to that of extracted vertebrate γTuRCs, even though individual yeast γTuSC protomers do not assemble into a higher-order defined ring in solution[16,17,19]. The geometry of the active γTuRC matches that of a 13-PF MT lattice perfectly, with a helical rise of 84 Å, a helical twist of ~28° and a width of 220 Å (Fig. 2a). Overall, our active γTuRC structure exhibits a substantially different geometry compared to the previously reported open structure of the recombinant yeast γTuSC filament[17,18], and other structures of inactive vertebrate γTuRCs[10–12] (Fig. 2b).

Relative to unassembled yeast γTuSC modules, and γTuSC modules assembled into the open γTuSC filament[17,18], the closed ring conformation of native γTuRC requires tightening and tilting of the two γ-tubulin subunits of each γTuSC module, so as to match the lateral α/β-tubulin geometry of the MT lattice (Fig. 2c). This γ-tubulin interface reorganization is facilitated by straightening of the gamma ring protein 2 (GRIP2) domain of Spc98 and bending of the GRIP2 domain of Spc97 (Fig. 2d). γTuRC ring closure is then accommodated by leaving an empty gap above the Spc97:γ-tubulin spoke 1 of γTuSC-1 (Extended Data Fig. 2b). Comparing our structure to human γTuRC, we observed that the first four GCP2/3 modules of human γTuRC match a 13-PF MT lattice, but the GCP4/5:GCP4/6 nucleation core as well as the last GCP2/3 module have to move inwards by ~50 Å to match the MT lattice geometry (Fig. 2b)[11]. The closed geometry of native γTuRC is similar but not identical to that of a recombinant γTuSC filament trapped in a closed state by engineered disulfides in γ-tubulin[18]. Ring closure induced by the γ-tubulin disulfide enhanced MT nucleation efficiency in vitro[17], a finding in support of the idea that, by matching the geometry of a MT lattice, γTuRC templates MT nucleation.

Ring closure at the γTuRC seam, which aligns with the MT lattice seam, is achieved through interactions between Spc98:γ-tubulin of γTuSC-7 and the first α/β/α-tubulin row of the MT lattice (Fig. 2e). The β-tubulin:Spc98 interface (Fig. 2e, ii) at the seam is formed mainly through salt bridge interactions between a basic Spc98^GRIP2 patch and acidic peripheral β-tubulin helices. The lateral α-tubulin:γ-tubulin interface at the γTuRC seam is also formed through electrostatic interactions (Fig. 2e, i) and reminiscent of the lateral α:β-tubulin interface at the MT lattice seam, with a single lateral contact site—a loop from the α-tubulin globular fold connecting α-tubulin and γ-tubulin. This is in contrast to two lateral contacts between α/β-tubulin molecules in the MT lattice, but similar to single-contact interactions between γ-tubulin molecules in native γTuRC (Extended Data Fig. 2c). Overall, the longitudinal γ:α-tubulin interface at the nucleated γTuRC is almost identical to the β:α-tubulin interface within the MT lattice (Extended Data Fig. 2c)[34]. At a resolution of 7 Å in this region, we cannot exclude that there are other subtle differences between the γ:α-tubulin and β:α-tubulin interfaces.

The essential Spc110 N-terminus forms a dimeric coiled coil (Spc-110^NCC) at the base of six γTuSC modules (γTuSC-1 to γTuSC-6) (Fig. 1c,d). The Spc110^NCC bound to the last γTuSC of the ring (γTuSC-7, spokes 13 and 14) also binds to the first γ-tubulin of γTuSC-1 (spoke 1), probably further stabilizing γTuRC ring closure (Fig. 1d). For all γTuSC modules, Spc110 then threads over the surface of an Spc98–Spc97 protomer and links adjacent γTuSC modules through the highly conserved CM1 motif (Spc110^CM1) that binds across the GRIP2 domains at each inter-γTuSC interface[18] (Extended Data Fig. 3a). Our reconstruction of the native γTuRC and its interactions with Spc110 matches a higher-resolution single-particle reconstruction of γTuSC filaments bound to Spc110 (ref. 18). In addition to the known interaction interfaces, at lower

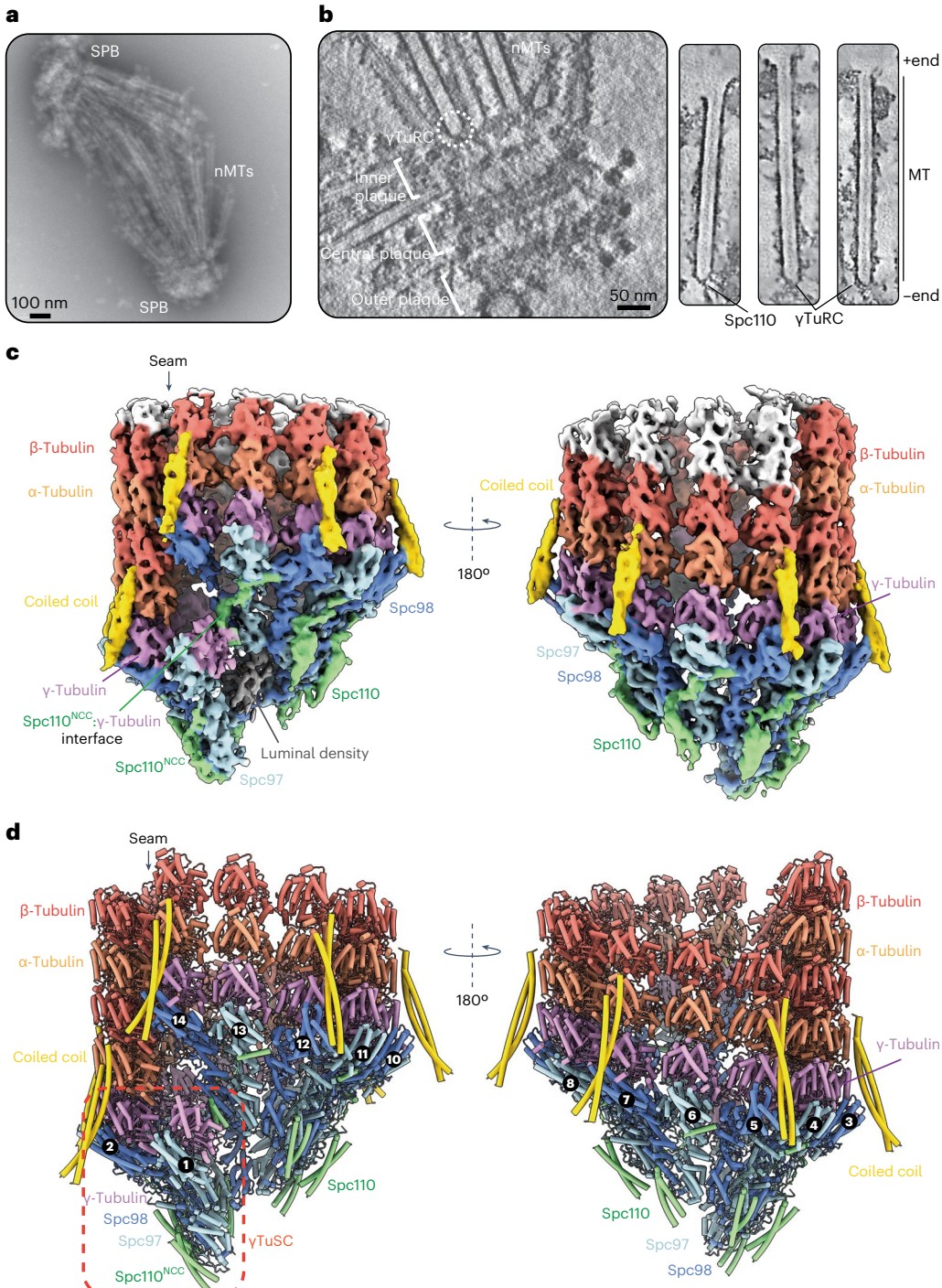

**Fig. 1 | Global architecture of the native budding yeast γTuRC capping MT minus ends. a**, A negative-stain electron micrograph of the *S. cerevisiae* SPBs forming a bipolar spindle, a typical structure observed following the enrichment procedure. Similar spindle structures were observed in more than ten independent SPB enrichment preparations. **b**, Left: a slice through a representative tomogram showing outer, central and inner plaques of the intact SPB, and a capped MT minus end (circled). Right: slices through denoised tomographic reconstructions of spindle MTs. MT-capping γTuRCs and Spc110 coiled coils connect the MTs to the SPB. The MT lattice and γTuRC caps are intact. **c**, A consensus cryo-ET reconstruction of the γTuRC capping the α/β-tubulin lattice after equalizing density thresholds using Occupy[52]. Scale bar as in **b**. **d**, A molecular model of the γTuRC bound to the MT minus end. Spokes are numbered as in Extended Data Fig. 2a, and a γTuSC unit is highlighted in a red box.

isosurface thresholds, we also observe Spc110^NCC of γTuSC-1 to γTuSC-6 converging into a single filament at the base of the γTuRC, consistent with Spc110 oligomerization being an important step during yeast γTuRC formation[20,21] (Extended Data Fig. 3b).

Three-dimensional (3D) classification of the MT subtomogram volumes against 11–16 PF reference structures revealed that the native MT lattice contains almost exclusively 13 PFs (97%, Extended Data Fig. 3c) consistent with γTuRC presenting 13 γ-tubulin molecules for MT nucleation. This is in contrast with yeast MTs nucleated in vitro using recombinant α/β-tubulin, where a highly variable PF number is observed, with 14 PFs being the most abundant[32], although in agreement with native yeast α/β-tubulin MTs nucleation in vitro[35].

**Table 1 | Cryo-ET data collection, refinement and validation statistics**

| | γTuRC<br>(EMD-18665), (PDB 8QV2) | γTuSC^subboxed<br>(EMD-18666), (PDB 8QV3) | MTs<br>(EMD-18664), (PDB 8QV0) |
|---|---|---|---|
| **Data collection and processing** | | | |
| Magnification | 81,000× | 81,000× | 81,000× |
| Voltage (kV) | 300 | 300 | 300 |
| Electron exposure (e⁻ Å⁻²) | 120 | 120 | 120 |
| Electron dose per tilt image (e⁻ Å⁻²) | 3.0 | 3.0 | 3.0 |
| Energy filter slit width (eV) | 20 | 20 | 20 |
| Tilt range (degrees) | +60 to −60 | +60 to −60 | +60 to −60 |
| Acquisition scheme | Dose-symmetric tilt scheme | Dose-symmetric tilt scheme | Dose-symmetric tilt scheme |
| Defocus range (µm) | 2–4.5 | 2–4.5 | 2–4.5 |
| Pixel size (Å) | 1.1 | 1.1 | 1.1 |
| Symmetry imposed | C1 | C1 | C1 |
| Initial subtomograms (no.) | 7,910 | 31,720 | 130,704 |
| Final subtomograms (no.) | 7,910 | 31,720 | 130,704 |
| Map resolution (Å) | 9.2 | 8.2 | 6.6 |
|     FSC threshold | 0.143 | 0.143 | 0.143 |
| Map resolution range (Å) | 8.1–13.4 | 7.1–8.8 | 6.6–7.5 |
| **Refinement** | | | |
| Initial model used (PDB code) | 7M2X | 7M2X, 5W3F | 5W3 |
| Model resolution (Å) | 10.8 | 9.1 | 8.2 |
|     FSC threshold | 0.5 | 0.5 | 0.5 |
| Model resolution range (Å) | 8.1–13.4 | 7.1–8.8 | 6.6–7.5 |
| Map sharpening *B* factor (Å²) | −155 | −115 | −105 |
| Model composition | | | |
|     Nonhydrogen atoms | 255,260 | 33,821 | 88,179 |
|     Protein residues | 32,221 | 4,267 | 11,271 |
|     Ligands | – | – | – |
| *B* factors (Å²) | | | |
|     Protein | 658.42 | 121.99 | 155.87 |
|     Ligand | – | – | – |
| Root mean square deviations | | | |
|     Bond lengths (Å) | 0.005 | 0.002 | 0.002 |
|     Bond angles (°) | 0.467 | 0.481 | 0.713 |
| Validation | | | |
|     MolProbity score | 1.71 | 1.79 | 2.02 |
|     Clashscore | 10.12 | 14.09 | 13.85 |
|     Poor rotamers (%) | 0.01 | 0.00 | 0.12 |
| Ramachandran plot | | | |
|     Favored (%) | 96.89 | 97.30 | 94.66 |
|     Allowed (%) | 3.06 | 2.67 | 5.22 |
|     Disallowed (%) | 0.05 | 0.02 | 0.12 |

## Coiled-coil proteins stabilize the first row of α/β-tubulin

The consensus reconstruction and the subboxed map revealed repetitive densities for a coiled-coil protein that spans each Spc98:γ-tubulin pair and the first row of α/β-tubulin (Figs. 1c and 3a,b). These are absent from all previously reported vertebrate and yeast γTuRC structures. The coiled-coil protein binds the Spc98 GRIP2 domain, γ-tubulin and α-tubulin, where the coiled-coil end flares: the density for one of the α-helices terminates while the other α-helix continues and interacts with the β-tubulin. The interface between the coiled coil and γTuSC:α/β-tubulin is shallow and is formed primarily by numerous highly conserved acidic residues on the γ-tubulin (Fig. 3b) and α-tubulin surfaces. The coiled coils appear to stabilize α:β-tubulin dimers at Spc98-bound γ-tubulin, effectively positioning the first α:β-tubulin dimers of the MT lattice. Notably, no coiled coils are bound across the Spc97:γ-tubulin:α-tubulin surface. A coiled coil bound to Spc97 would probably promote α:β-tubulin binding to the

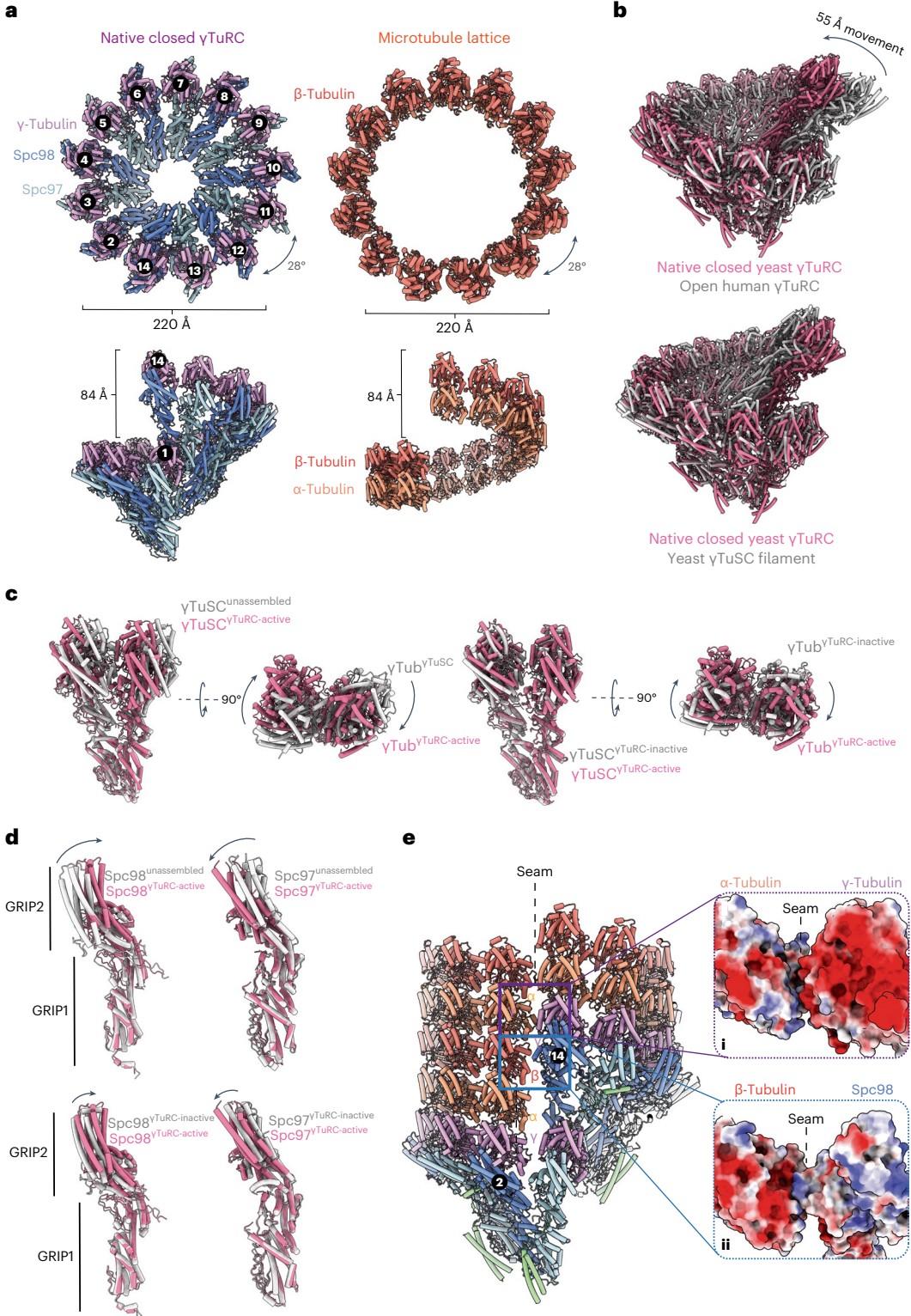

**Fig. 2 | γTuRC forms a perfect template for MT nucleation. a**, Top-down and side views of the native yeast γTuRC core complex (left) and the MT lattice (right) showing an identical helical geometry. The MT lattice was modeled on the basis of our 6.6 Å reconstruction of the native yeast MT lattice (Extended Data Fig. 1a). '(b)' refers to the view shown in **b. b**, The native, closed and active yeast γTuRC structure was superimposed onto the inactive human γTuRC complex (PDB ID: 6V6S) (top) and single helical turn of the yeast γTuSC filament (PDB ID: 7m2x) (bottom), showing that γTuRC undergoes extensive conformational changes to match MT geometry and thus template MT nucleation. **c**, Left: a comparison of γTuSC geometries between unassembled γTuSC modules (PDB ID: 7m2z)

and γTuSC modules in the context of active γTuRC. Right: a comparison of γTuSC geometries between γTuSC within the open-inactive γTuSC filament (PDB ID: 7m2x) and γTuSC in the context of active γTuRC. γ-tubulin pairs of each γTuSC module straighten up to adopt 13-PF MT geometry. Coordinates were superimposed on the Spc98^GRIP1 domain. **d**, γ-Tubulin pairs straightening is accommodated by Spc98^GRIP2 straightening and Spc97^GRIP2 bending. PDBs were superimposed on the Spc98^GRIP1 domain. **e**, A view of the interfaces at the γTuRC:MT seam, focusing on the electrostatic α-tubulin:γ-tubulin as well as Spc98:β-tubulin interfaces. The insets on the right show the α-tubulin:γ-tubulin (i) and Spc98:β-tubulin (ii) interfaces as electrostatic surfaces.

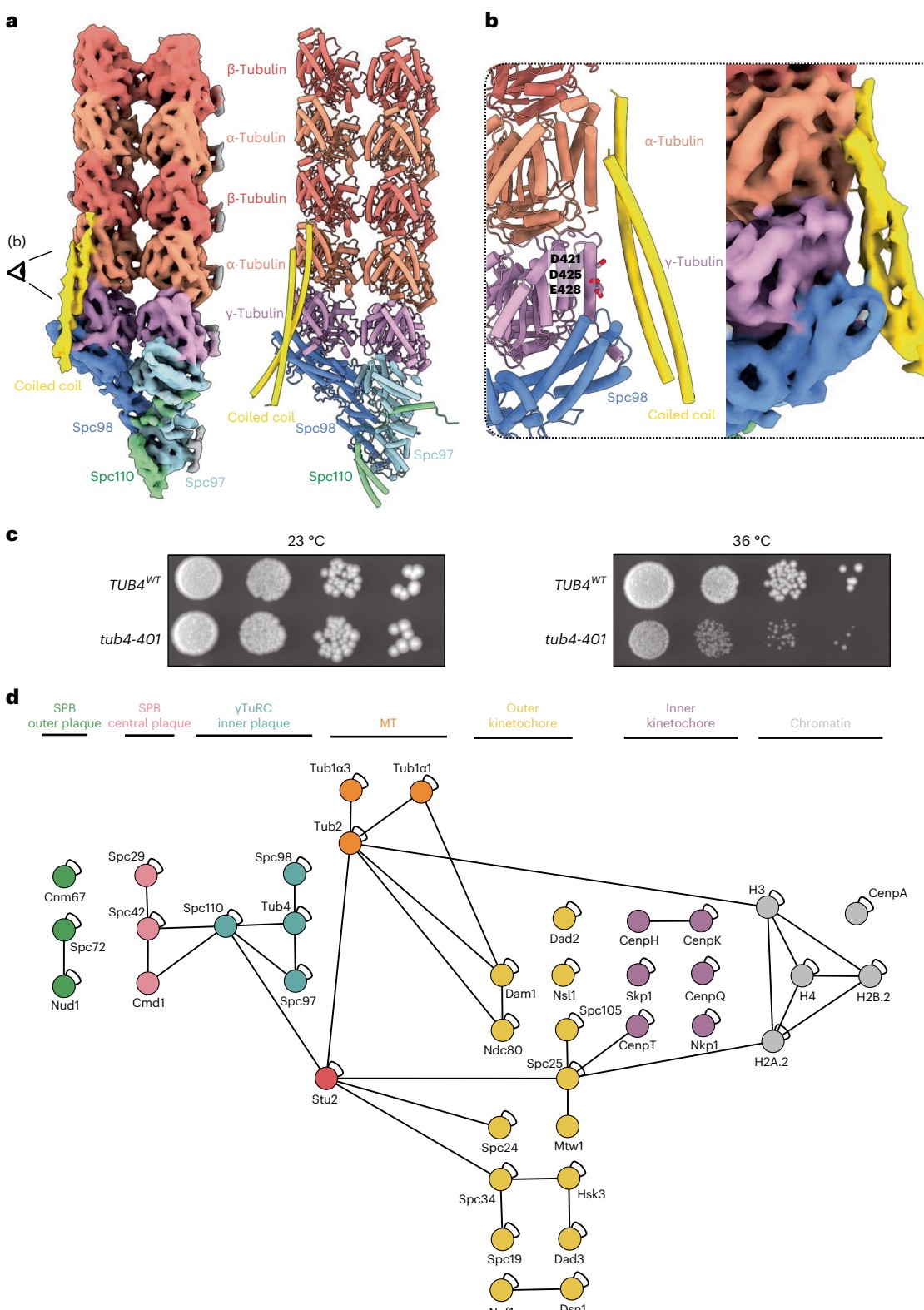

**Fig. 3 | A coiled-coil protein spans the Spc98:γ-tubulin:α-tubulin interface.**
**a**, A subboxed cryo-ET map at 8 Å, and molecular model of a γTuSC bound to the first two rows of α/β-tubulin. **b**, A model and map of the coiled-coil interaction with Spc98, γ-tubulin and α/β-tubulin. The crystal structure of the coiled-coil domain of Stu2 (ref. 38) fits the coiled-coil density in the subtomogram average, as shown in **b**. **c**, The *tub4-401* mutant (γ-tubulin mutant D421R, E425R and E428R), predicted to disrupt the coiled-coil interaction with γ-tubulin, grows normally at 23 °C but shows growth defects at 36 °C. TUB4^WT, wild-type γ-tubulin. **d**, Intra- and intermolecular cross-links observed during CLMS experiments of

the enriched SPB preparation. Proteins are shown as nodes, labeled with their respective protein names, and color-coded according to their protein group affiliation (as indicated above). Cross-links identified from the listed protein groups through CLMS analysis of the enriched SPB preparation are depicted. Self-links are represented by loops on the nodes, while heteromeric cross-links are denoted by lines connecting the nodes. γTuRC proteins are labeled in dark cyan. Two coiled-coil proteins, Stu2 and BRE1, and one globular protein (Ima1) were found to be cross-linked to γTuRC proteins, of which Stu2, known to be enriched at SPBs, is shown in red.

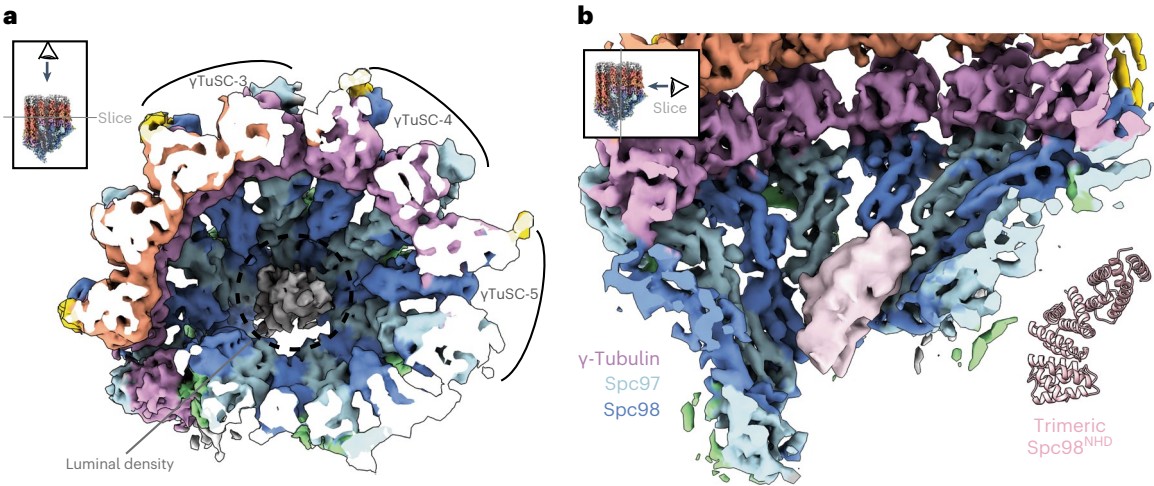

**Fig. 4 | The γTuRC luminal density. a**, A top-down view of the γTuRC cryo-ET map showing an elongated luminal bridge density at the center of the γTuRC structure. **b**, A view of luminal density (pink, low-pass filtered) and the AlphaFold2 prediction of a trimeric Spc98^NHD.

first Spc97:γ-tubulin spoke (Extended Data Fig. 2b, spoke 1), which would interfere with γTuRC ring closure, potentially explaining why coiled-coil binding to Spc97 could be biologically undesirable.

The observation that the coiled-coil protein binds exclusively to Spc98:γ-tubulin:α-tubulin, and not to Spc97:γ-tubulin:α-tubulin, implies there are Spc98-specific structural features that underlie this specificity. First, whereas a basic patch on the Spc98 GRIP2 domain faces the coiled-coil protein, an acidic patch is exposed on the corresponding GRIP2 domain of Spc97 (Extended Data Fig. 4a). Second, Spc97^GRIP2 exposes a large unstructured loop (residues 720–753, 'GRIP2^Loop' (Extended Data Fig. 4b), absent in Spc98, which might prevent the coiled coil from binding. Last, the acidic carboxy-terminal region of γ-tubulin (γ-tubulin^CTR) interacts differently with the Spc97^GRIP2 and Spc98^GRIP2 domains in our γTuRC reconstruction. Alphafold2 modeling indicated that the conserved 'DSYL' motif of γ-tubulin^CTR forms a small interface with Spc97^GRIP2 and Spc98^GRIP2, a prediction supported by high-resolution structures of γTuSC[18] (Extended Data Fig. 4b,c). C-terminal of the 'DSYL' motif, γ-tubulin^CTR is predicted to form more extensive interactions with the Spc98^GRIP2 residues (but not with Spc97^GRIP2) through hydrophobic interactions. These Spc98-specific interfaces would direct γ-tubulin^CTR toward the coiled-coil density (Extended Data Fig. 4d). These predictions are supported by cryo-ET density spanning Spc98^GRIP2 in our subtomogram average (Extended Data Fig. 4b–d). Overall, Spc98 presents specific structural features that could confer specificity for coiled-coil binding.

To investigate the function of the coiled-coil protein, we tested if the coiled-coil interaction with the γTuRC is functionally important. We reversed the charge of a conserved acidic patch on γ-tubulin (γ-tubulin D421R, E425R and E428R) that directly engages the coiled coil and that is the largest interface with the coiled-coil density (Fig. 3b). We observed a temperature-sensitive phenotype in a yeast strain with a mutation in the γ-tubulin gene (*tub4-401*), indicating that the γ-tubulin:coiled-coil interface is important for cellular function (Fig. 3c). Due to likely pleiotropic effects, no mutants of α/β-tubulin were tested. This result indicated that the interaction between the coiled-coil staple and γ-tubulin:Spc98 is functionally important for MT formation and/or stability.

Due to the limited resolution, it was not possible to establish the identity of the coiled-coil protein from the cryo-ET maps. To define candidate proteins that could account for the coiled-coil density, we subjected our enriched SPB sample to mass spectrometry (MS) (Supplementary Data 1). Consistent with our structural work, α- and β-tubulin were the most enriched proteins, with abundant SPB components such as Spc110 and other components of the γTuRC also enriched (Extended Data Fig. 5a). We then focused on proteins that localize to

**Table 2 | X-ray data collection and refinement statistics**

| Protein name, PDB ID | Spc98^NHD, (PDB:8QRY) |
|---|---|
| **Data collection** | |
| Space group | P2₁2₁2₁ |
| Cell dimensions | |
| *a, b, c* (Å) | 28.3, 45.3, 65.6 |
| α, β, γ (°) | 90.0, 90.0, 90.0 |
| Resolution (Å) | 65.6-1.87 (1.9–1.87)^a |
| $R_{sym}$ | 0.10 (0.35) |
| *I/σI* | 8.7 (0.9) |
| Completeness (%) | 99.2 (78.8) |
| Redundancy | 3.3 (2.5) |
| **Refinement** | |
| Resolution (Å) | 37.3-1.87 |
| Number of reflections | 7,294 |
| $R_{work}/R_{free}$ | 0.24/0.30 |
| Number of atoms | |
| Protein | 762 |
| Ligand/ion | – |
| Waters | 44 |
| *B* factors (Å²) | |
| Protein | 21.5 |
| Water | 23.5 |
| Root mean square deviations | |
| Bond lengths (Å) | 0.008 |
| Bond angles (°) | 1.045 |

One crystal was used in this dataset. ^aValues in parentheses are for highest-resolution shell.

the spindle MTs and contain a coiled coil of >10 nm. Based on these criteria Stu1, Bim1 (binding to MTs 1) and a few other proteins could be excluded, since they either lack a predicted coiled coil or have coiled coils that are too short to occupy the coiled-coil density in our reconstruction. However, most other known MT-associated proteins (MAPs) such as the kinesins Kip1 and Cin8, the MT polymerase Stu2 (vertebrate XMAP215 ortholog), Bik1 and many components of the kinetochore are present in the enriched SPB sample, are known to localize close to the

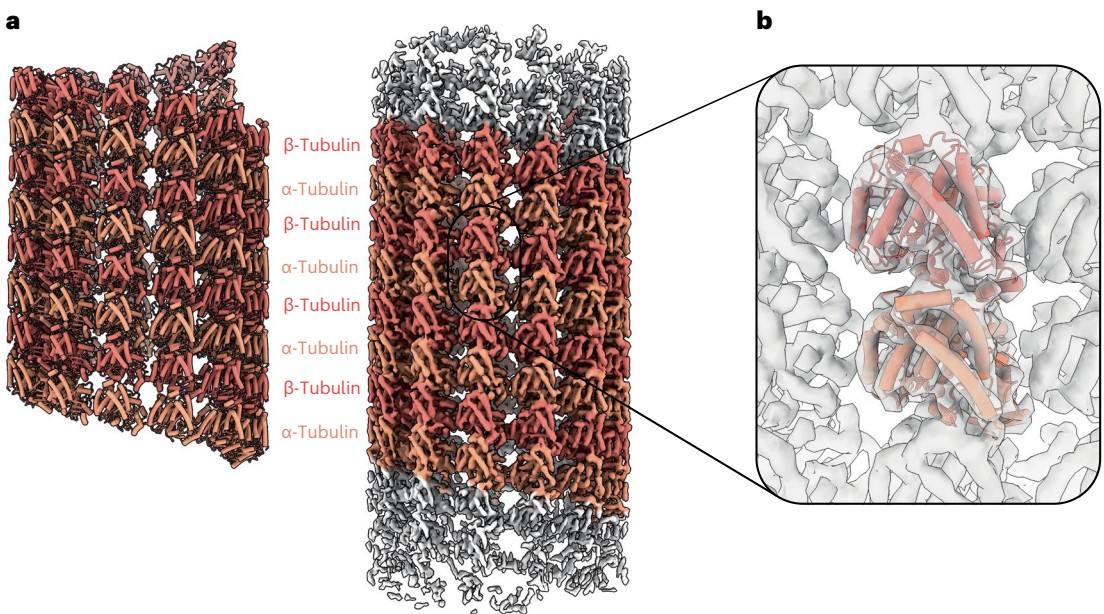

**Fig. 5 | MT consensus reconstruction and α/β-tubulin lattice model.**
**a**, A model and 6.6 Å cryo-ET reconstruction of native yeast spindle MTs.
Measurement of the distance between adjacent α/β-tubulins shows that the
MT lattice is in the compacted state, when compared to MT lattice structures
determined in vitro[32]. **b**, A focused view on the individual α/β-tubulin unit with
molecular model overlayed with the reconstructed cryo-ET map.

SPBs in cells and have a coiled-coil domain of at least 10 nm. Spc110 is
also a possible candidate; it contains coiled-coil regions essential for
cellular function, albeit none N-terminal of the CM1 helix.

To further define the coiled-coil protein, we performed cross-linking mass spectrometry (CLMS) analysis of our enriched SPB preparation, which revealed many cross-links that are consistent with known interactions of the SPB plaques, γTuRC and kinetochore (Fig. 3d)[16,29,36]. Apart from known components of the γTuRC, CLMS did not detect any other proteins interacting directly with either Spc98 or γ-tubulin. However, of three newly identified cross-linked proteins to Spc110, two are coiled-coil proteins: Bre1 and Stu2. Furthermore, of these two proteins, only Stu2 is enriched at SPBs, cross-linking near the Spc110 N-terminus, although not via its coiled coil (Fig. 3d and Extended Data Fig. 5a,b). Stu2 is an essential protein in yeast that co-localizes with γ-tubulin in cells, even in the absence of MTs, suggesting that its interaction with γTuRC might be independent of α:β-tubulin[37]. Stu2 has a parallel, homodimeric 15 nm coiled coil, which is important for its function in cells[38], as well as two tumor overexpressed gene (TOG) domains that bind free α/β-tubulin and stimulate MT polymerization[39,40]. In follow-up experiments, we were unable to detect meaningful cross-links between recombinant Stu2 and recombinant budding yeast γTuRC (composed of Spc97, Spc98, Tub4 and Spc110 amino acids 1–400, but without yeast tubulin) using CLMS. In addition, we could not detect an interaction between recombinant γTuRC and Stu2 using gel-filtration reconstitution experiments. Overall, we attempted to narrow down the list of potential coiled-coil candidates, and found a plausible hit, but its precise identity remains unknown and requires further investigation.

## γTuRC luminal density bridges γTuSC protomers

In addition to the coiled-coil densities, we observe an additional density feature in the lumen of native γTuRC, unaccounted for by existing yeast γTuRC models, but reminiscent of the luminal bridge (γTuRC[LB]) described in higher eukaryotes[41,42] (Figs. 1c and 4a,b). The γTuRC luminal density is compact and connects multiple γTuSC modules from the inside of the lumen, interacting specifically across inter-γTuSCs interfaces of γTuSC-3 to γTuSC-5. Deletion of the γTuRC[LB] from higher eukaryotes results in incomplete γTuRC assembly, suggesting that it helps stabilize γTuRC oligomerization[13,43], while Mzt1 (a component of the γTuRC[LB]) has been shown to regulate γTuRC-mediated MT nucleation[43].

We do not observe density for actin, present in vertebrate γTuRC[10–12], in the lumen of the *S. cerevisiae* γTuRC. This is consistent with actin not being present in the *S. cerevisiae* nucleus[44].

The γTuRC[LB] in vertebrate γTuRC is formed by Mzt1 intertwined with either the GCP3 or GCP6 (Spc98 homologs) N-terminal helical domains (GCP3[NHD] and GCP6[NHD])[41,42,45] (Extended Data Fig. 6a). However, the *mzt1* gene is absent from *S. cerevisiae*, and we did not detect other small α-helical proteins in our MS analysis (Extended Data Fig. 5a and Supplementary Data 1). *S. cerevisiae* Spc98 has a structured N-terminal α-helical domain (Spc98[NHD]), as predicted by AlphaFold2, not visible in the inactive recombinant yeast γTuSC filaments. We determined a crystal structure of the Spc98[NHD] at 1.9 Å resolution (Table 2), revealing a compact α-helical bundle nearly identical to the AlphaFold2 prediction (Fig. 4b). Interestingly, *S. cerevisiae* Spc98[NHD] is more compact than the extended GCP3[NHD]/GCP6[NHD] folds alone and thus more similar in size and fold to half of the composite Mzt1-GCP3[NHD]/GCP6[NHD] structures (Extended Data Fig. 6a). This observation could explain the absence of Mzt1 in budding yeast as Spc98[NHD] is more compact and its structure by itself makes up for the lack of Mzt1. The γTuRC luminal bridge density can in principle accommodate a trimer of the Spc98[NHD] crystal structure, in agreement with an AlphaFold2 prediction of a trimeric Spc98[NHD] oligomer (head-to-tail) (Fig. 4b and Extended Data Fig. 6b). Additionally, we observe densities connecting the N-terminal base of Spc98 and the luminal γTuRC density (Extended Data Fig. 6c), consistent with our preliminary annotation of the luminal bridge as a trimer of Spc98[NHD]. It is likely that the other four Spc98[NHD] domains are not bound in the γTuRC lumen, and are averaged out due to the flexible linker between Spc98[GRIP1] and Spc98[NHD], similar to γTuRC in higher eukaryotes[10–12]. Consistent with our model, deletion of Spc98[NHD] in *S. cerevisiae* impaired cell growth under stress and reduced γTuRC oligomerization propensity in vitro[21]. Overall, we show that the luminal bridge is a feature of all reported native γTuRC structures, probably facilitating formation of the complete γTuRC architecture, and that the luminal bridge remains associated with the active γTuRC at capped MT minus ends at the budding yeast γTuRC.

## The endogenous MT lattice is in the compacted state

We reconstructed the native yeast MT lattice to 6.6 Å resolution (Fig. 5a,b and Extended Data Fig. 1b). The MT lattice in our tomograms is densely decorated with MAPs that could not be resolved during

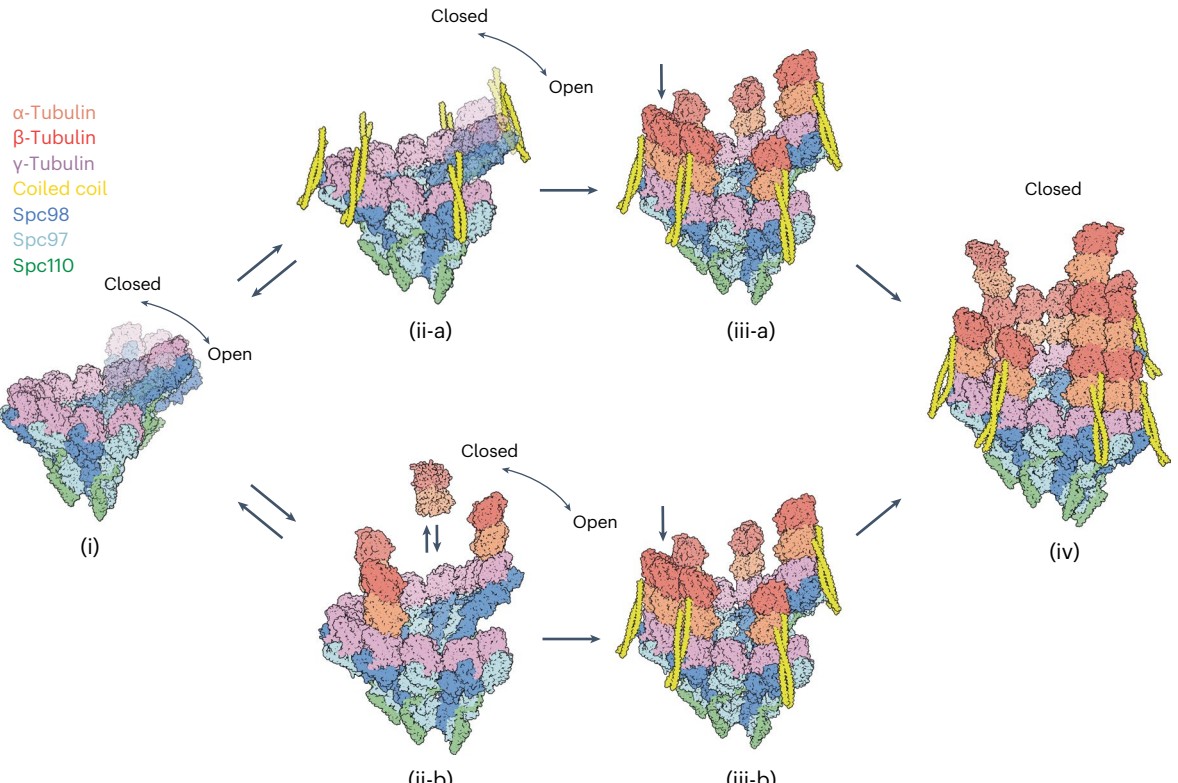

**Fig. 6 | Proposed molecular models of γTuRC activation and MT nucleation.** Spc110 recruits γTuSC modules to the SPB, where γTuSCs oligomerize into partial or complete γTuRCs assisted by the SPC110^CM1 motif and the luminal bridge density. Before MT nucleation and lattice formation, γTuRC cycles between open and closed states (i). Top: the coiled-coil protein engages Spc98 and γ-tubulin (ii-a) and subsequently helps recruit and/or stabilize the first row of α/β-tubulin at the γTuRC (iii-a). Recruited and stabilized α/β-tubulin dimers extend longitudinally and fill up high-affinity gaps laterally (gray arrow iii-a), forming tapered MT ends suited for efficient MT polymerization and growth (iv). An alternative mechanism of nucleation is possible, shown at the bottom:

α/β-tubulin dimers bind transiently to the γ-tubulin ring of open/closed γTuRC complexes (ii-b). The coiled-coil proteins bind to the extended Spc98: γ-tubulin: α-tubulin interfaces and stabilize the transient lateral interactions (iii-b). Recruited and stabilized α/β-tubulin dimers extend longitudinally and fill up high-affinity gaps laterally (gray arrow), forming tapered MT ends suited for efficient MT polymerization and growth (iv). In a third scenario (not depicted), the coiled coil binds to α/β-tubulin dimers before engaging γTuRC, as such stabilizing the weak lateral interactions. Given the molar excess of free α/β-tubulin in the yeast nucleus, this pathway is not likely to occur in vivo.

consensus map refinement due to conformational heterogeneity. The native yeast MT lattice is in a partially compacted state compared to structures of yeast MT reconstituted in vitro[32], suggesting that the native nucleotide state as well as MAPs result in a well-defined MT lattice spacing in cells. The resolution of our reconstruction did not permit the unambiguous assignment of the nucleotide state of α/β-tubulin.

## Discussion

Since the discovery of γTuRC as the universal MT nucleator almost 30 years ago[6,46], the precise molecular mechanism of γTuRC-mediated MT nucleation has remained elusive. We describe a 7–9 Å resolution structure of a native γTuRC capping nMT minus ends at the MTOC, obtained through subtomogram averaging. We observe γTuRC in a closed conformation, with conformational changes of the GRIP2 domains in each of the γTuRC spokes, presenting 13 γ-tubulin molecules as a perfect geometric template for MT nucleation. Recent studies of active human γTuRC bound to the MT lattice illustrate that this geometric templating mechanism is conserved between human and yeast cells[47,48]. Arresting features of our native γTuRC structure are the long coiled-coil proteins bound to every Spc98:γ-tubulin:α-tubulin interface, as well as the luminal bridge density.

We observe that the luminal density in our native budding yeast γTuRC reconstruction is reminiscent of the Mzt1:GCP3/6^NHD luminal bridge found in higher eukaryotes. This observation indicates that

the luminal bridge is a feature of all reported native γTuRC structures, probably facilitating formation of the complete γTuRC architecture. In budding yeast, however, there is no Mzt1 homolog, nor nuclear actin, and we propose that the luminal density corresponds to an oligomer of the compact Spc98^NHD helical bundle (Fig. 4a). Whereas we observe the luminal density in the native γTuRC capping MT minus ends, the luminal bridge, including actin, is present in the open conformation of the human γTuRC but appears to be displaced from the fully closed human γTuRC after MT nucleation[48]. While the more substantial Mzt1:GCP3/6^NHD:actin luminal bridge was proposed to sterically hinder human γTuRC closure upon MT nucleation, the more compact luminal density in budding yeast γTuRC can be accommodated upon closure, as seen in our reconstructions.

At least two functions can be envisaged for the coiled-coil protein. (1) The coiled coil might stabilize attachment of the nMT minus end to γTuRC to prevent nMT detachment and dissociation throughout the cell cycle. Unlike vertebrate MTOCs, yeast γTuRC stably caps MT minus ends at the SPB during most of the cell cycle. (2) The coiled-coil protein might be involved in the process of MT nucleation by directly recruiting the first row of α/β-tubulin to γTuRC. In vitro, γTuRC activation and MT nucleation from γTuRC requires either four[25] or seven[12] α/β-tubulin heterodimers bound to the γ-tubulin ring. It is plausible that the coiled-coil protein facilitates this step by stabilizing up to seven α/β-tubulin heterodimers longitudinally (one for every Spc98:γ-tubulin spoke). Importantly, these two mechanisms of action are not mutually exclusive.

Unambiguous identification of the coiled-coil protein in future studies will allow dissection of its function and will also provide insights into whether similar mechanisms exist in other organisms. Whether the coiled-coil domains are bound to and function at the cytoplasmic γTuRC remains an open question. Lastly, whether 'coiled-coil' proteins stabilize longitudinal interactions of the first row of α/β-tubulin in native higher eukaryote γTuRC remains to be discovered. In reconstituted human γTuRC, an unidentified 'latch' was observed at the MT base, laterally bridging γ-tubulin at spoke 1 and β-tubulin at spoke 14 at the seam base of a 'closing' γTuRC intermediate. This observation indicates that additional regulatory components play a role in human γTuRC activation and MT nucleation[48]. Interestingly, the latch density is no longer observed in reconstructions of fully closed human γTuRC, and no such density was observed in our reconstructions of closed budding yeast γTuRC either. In our reconstruction, γTuRC closure at the seam base is stabilized solely by electrostatic interfaces between Spc98[GRIP2]: β-tubulin and γ-tubulin:α-tubulin (Fig. 2e).

Our structural findings together with the wealth of prior functional and biochemical data allow us to propose two plausible pathways of MT nucleation by yeast γTuRC (Fig. 6). Spc110 recruits individual γTuSCs to the SPB, which facilitates their oligomerization into a complete γTuRC, an activity that is promoted by Spc98[NHD], forming the luminal bridge, as well as the Spc110[CM1] helix, stabilizing adjacent γTuSCs specifically at the SPB (Fig. 6, i)[18]. Fully or partially assembled γTuRC can recruit the coiled-coil protein, while in an equilibrium of open and closed conformations (Fig. 6, inset ii-a), or transiently bind α/β-tubulin though longitudinal interactions with γ-tubulin (Fig. 6, inset ii-b). γTuRC together with the coiled-coil protein then stabilizes attachment of up to seven α/β-tubulin dimers on top of the central ring scaffold (Fig. 6, inset iii-a/b). These longitudinal attachments would normally dissociate in the absence of additional stabilizing factors, as shown in an in vitro reconstituted system where a high local concentration of α/β-tubulins bound to γTuRC was necessary to achieve nucleation[25]. In the native yeast system, however, the coiled-coil domains would stabilize the initial α/β-tubulin binding to the γTuRC. The alternating arrangement of the coiled-coil protein and recruited α/β-tubulin dimers present high-affinity gaps for additional α/β-tubulin dimers to fill, promoting intrinsically weak lateral α/β:α/β interactions and building up the first MT lattice row (Fig. 6, inset iii-a/b gray arrow, inset iv)[49,50]. Other than the Spc98[NHD] luminal bridge that links γTuSCs 3–5, together with Spc110[CM1] (Fig. 4a,b), we observe no other factors that could laterally activate γTuRC into a closed state to match the MT lattice geometry. Moreover, γTuSC-1 in our native γTuRC structure, which has no α/β-tubulin bound to the first γ-tubulin (spoke 1), is in the open state and does not match the MT lattice geometry, indicating that MT lattice formation is necessary for γTuRC activation. These observations are consistent with the hypothesis that initial MT lattice formation during nucleation itself locks γTuRC into a closed state[25]. This 'lattice-driven' activation of γTuRC through lateral α/β-tubulin interactions was also proposed in recent studies of reconstituted human γTuRCs capping MT minus ends[47,48]. The initial α/β-tubulins at the γ-tubulin ring, positioned by the coiled-coil protein, would resemble tapered MT ends, which have been proposed to be a key structural intermediate of the growing MT tip[5,24,51] (Fig. 6, inset iv). Overall, γTuRC, together with the coiled-coil protein recruiting/stabilizing α/β-tubulins, generates a stable scaffold that functions as a structural mimic of a native polymerizing MT plus end.

## Online content

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

## Methods

### Preparation of the *S. cerevisiae* nuclei

*S. cerevisiae* NCYC74 nuclei were prepared as described in ref. 29. This yeast strain generates a high yield of SPBs.

### SPB enrichment

SPBs were prepared from *S. cerevisiae* nuclei[53] up to the sucrose equilibrium gradient step. For structural work, enriched SPBs were first dialyzed against bt-buffer (10 mM bis-Tris/HCl pH 6.5 and 0.1 mM MgCl$_2$) supplemented with 20% dimethyl sulfoxide (DMSO) solution and then dialyzed once more against bt-buffer supplemented with 5% DMSO in a new dialysis cap. The final enriched SPBs were used for cryo-electron microscopy (cryo-EM) grid preparation and cryo-ET.

### TUB4 mutagenesis and yeast strains

Mutations in TUB4 (termed *tub4-401*) were prepared by overlap extension PCR[54]. TUB4 constructs in a modified version of pRS 305K (ref. 55) containing a NotI site in the Kanamycin polylinker were cut with ApaLI and XhoI and integrated at LEU2 in a diploid strain containing a disruption of TUB4 with HIS3MX6. This strain, together with a wild type rescue, were sporulated and KAN+HIS+ spores retained.

### Protein expression and purification

Spc98$^{NHD}$ (amino acids 1–95) was cloned into the pRsfDuet vector with an N-terminal 6xHis-tag. Spc98$^{NHD}$ was expressed in BL21 Star (DE3) cells using 2xTY medium. Cells were grown at 37 °C and induced using isopropyl β-ᴅ-1-thiogalactopyranoside when the OD$_{600}$ reached 0.7–0.8, after which the temperature was lowered to 18 °C. The cell pellet was lysed by sonication in lysis buffer (50 mM Tris–HCl pH 8.0, 1 M NaCl, 10 mM imidazole and 5% glycerol) supplemented with benzamidine, EDTA-free protease inhibitor tablets, benzonase and lysozyme (Sigma-Aldrich). Clarified lysate was loaded onto HiTRAP TALON (GE Healthcare) columns, washed with ten column volumes and eluted using 300 mM imidazole in the lysis buffer. The 6xHis-tag was cleaved overnight using tobacco etch virus protease in a dialysis buffer containing 20 mM HEPES pH 7.8, 500 mM NaCl, 15 mM imidazole and 5% glycerol. Cleaved protein was reverse-passed through HiTRAP HP columns (Cytiva), and the flow-through protein was collected, concentrated to 50 mg ml$^{-1}$ and stored.

### Spc98$^{NHD}$ crystallization and structure determination

Purified Spc98$^{NHD}$ (50 mg ml$^{-1}$) was crystallized using the Morpheus Screen (Molecular Dimensions) in a sitting drop format. One-hundred nanoliters of the Spc98$^{NHD}$ protein was mixed with 100 nl of the reservoir solution that contained 0.12 M monosaccharides, 0.1 M Buffer System 1 pH 6.5 and 37.5 % v/v of Precipitant Mix 4. The drop was left for 3 days at 18 °C when crystals were collected. No additional cryoprotectant was applied. The crystal was directly mounted on the LMB in-house X-ray generator (Rigaku FrR-E+ SuperBright), with data collected at a temperature of 100 K with wavelength of 1.54 Å. Data were indexed and integrated with MOSFLM[56] and scaled and merged with AIMLESS[57]. Phasing was achieved by molecular replacement with Phaser[58] as implemented in PHENIX, using a model generated by AlphaFold2 (ref. 59).

### Spc98$^{NHD}$ structure refinement

The structure was refined using phenix.refine[60]. Ramachandran statistics were 95.74% preferred and 4.26% allowed.

### Negative-stain electron microscopy

Three microliters of sample (SPBs or protein at 0.05 mg ml$^{-1}$) were applied to glow-discharged (Edwards S150B glow discharger, 40 s) carbon supported 400 Cu grid (Agar Scientific) and allowed to adhere for 30 s. After that, uranyl acetate (UA) solution was applied to the sample, which was simultaneously blotted away using Whatmann blotting paper until 2 µl of UA solution was left on the grid. After 30 s of staining, the UA solution removed by blotting and the grid was left to dry.

### SPB cryo-ET data collection

Three microliters of enriched SPBs mixed with bovine serum albumin-coated 10 nm gold fiducials (BBI Solutions) were applied to Lacey Carbon 300 mesh Au grids (Agar Scientific) either with a continuous carbon film or without any additional support, glow discharged using an Edwards S150B glow discharger (1 min for grids without additional carbon layer, or 30 s for grids with the additional carbon support layer), setting 6, 30–35 mA, 1.2 kV, 0.2 mBar (0.15 Torr). The grids were then flash frozen in liquid ethane using a ThermoFisherScientific Vitrobot IV (0.5 s waiting time, 2 s blotting time, −7 blotting force for grids without additional carbon support layer; 30 s waiting time, 5 s blotting time, 10 blotting force for grids with additional carbon support layer) using Whatman filter paper 1. Cryo-EM images were collected on a ThermoFisherScientific Titan Krios microscope operating at 300 keV using a Gatan K3 camera and a BioQuantum energy filter. Tilt series were collected using SerialEM at a nominal magnification of 81k, yielding a pixel size of 1.1 Å per pixel. The dose was set at 3.0 e$^-$ Å$^{-2}$ per tilt image collected over ten frames. Data collection was performed using a dose-symmetric tilt-series acquisition scheme with 3° increments at stage tilts between −60 and +60, giving a total of 41 tilt-series images. The total dose over the tilt series was 123 e$^-$ Å$^{-2}$.

### γTuRC and MT data processing

A detailed pipeline for data processing is provided in Extended Data Fig. 1. All tilt series were preprocessed (motion correction and CTF estimation) in WARP[61] and manually curated to exclude bad images from the tilt series. The dAutoalign4warp package was used to perform gold-fiducial-based tilt series alignments in dynamo[62]. Tomograms were generated in WARP at 8.8 Å per pixel and deconvolved for particle picking. Dynamo dtmslice was used to manually pick all γTuRC and MT particles[63]. γTuRC particles were picked as dipoles (z axis along the MT, away from the SPB) while MT particles were picked as filaments (z axis along the MT, away from the SPB). The coordinates (dynamo.tbl format) were converted to RELION 3.0 star files with the dynamo2m package. Subtomogram volumes and 3D CTF volumes were reconstructed at 6.6 Å per pixel for both the γTuRC and MT species using WARP. Initial 3D references were obtained in Dynamo by randomizing the azimuth of 1,000 γTuRC or MT particles and subsequently averaging them to obtain a featureless cone or barrel for the γTuRC and MT respectively. RELION 3.1 (refs. 64,65) was used to 3D to align the subtomogram volumes to γTuRC and MT 3D references. Three-dimensional classification in RELION did not reveal poor particles. The initial poses for γTuRC and the MTs were co-refined in M (ref. 66) at 3.3 Å per pixel. Tilt-series image warp, stage angles and defocus were also refined in M. γTuRC subtomogram volumes were reconstructed with M at 3.3 Å per pixel, and helical misalignments of the seam were resolved and corrected using RELION 3D classification and in-house scripts (fixalign), respectively. Napari subboxer[67] was used to expand the γTuRC dataset by subboxing and refining γTuSC subunits, improving the resolution substantially.

Three-dimensional classification of MTs was used to separate MT species according to PF numbers, following the MiRP procedure[68].

### Model building

The atomic model of yeast γTuSC[18] was fitted into a subboxed map of the γTuSC bound to two rows of α/β-tubulin and manually corrected in Coot[69]. The model for α/β-tubulin in the GDP state was fitted into the corresponding portion of the density and manually corrected. The coiled-coil density was built by positioning ideal α-helices into the coiled-coil density. The resulting model was subsequently refined with Phenix using standard settings. The model of the γTuSC was then fitted into one PF of the γTuRC, which was subsequently duplicated and repeated seven times to generate a complete model of the γTuRC,

docking additional α/β-tubulin dimers into the map. The final model was further refined in PHENIX[60] using standard settings.

## CLMS

Enriched SPBs from the sucrose equilibrium gradient[53] were dialyzed against 10 mM MES pH 6.5, 20% DMSO and 0.1 mM MgCl₂. Dialyzed SPBs (500 µl) were made 17 mM in BS3 and left in ice for 20 min, then quenched by addition of 25 µl 1 M Tris pH 8.0 for 20 min in ice. Cross-linking efficiency was tested through western blot, showing that the Spc98 band shifts after cross-linking. The sample was precipitated by addition of three volumes of acetone and dried. The resulting precipitated proteins were then solubilized in a digestion buffer containing 8 M urea in 100 mM ammonium bicarbonate, aiming for an estimated protein concentration of 1 mg ml⁻¹. Subsequently, the dissolved protein sample underwent reduction by the addition of 1 M dithiothreitol (DTT) to achieve a final concentration of 5 mM, followed by a 30 min incubation at 25 °C. The free sulfhydryl groups within the sample were alkylated by the addition of 500 mM iodoacetamide, resulting in a final concentration of 15 mM, and incubated at room temperature for 20 min in darkness. To neutralize any excess iodoacetamide, additional 1 M DTT was introduced, bringing the total concentration in the sample to 10 mM. The protein samples were then subjected to digestion with LysC at a 50:1 (m/m) protein to protease ratio, conducted at 25 °C for 6 h. Following this, the sample was diluted with 100 mM ammonium bicarbonate to achieve a urea concentration of 1.5 M, and trypsin was added at the same protein to protease ratio for further protein digestion overnight, approximately 15 h. The resulting peptides were desalted using C18 StageTips[70].

For protein identification, 0.5% of the total peptides underwent direct analysis by liquid chromatography–tandem mass spectrometry (LC–MS/MS) (referred to as the protein ID sample). The LC–MS/MS analysis was performed using a Q Exactive HF mass spectrometer (Thermo Fisher Scientific) connected to an Ultimate 3000 RSLCnano system (Dionex, Thermo Fisher Scientific). Peptides were resuspended in a solution containing 1.6% v/v acetonitrile and 0.1% v/v formic acid, then injected onto a 50 cm EASY-Spray C18 LC column (Thermo Scientific) operating at 50 °C column temperature. The mobile phase consisted of water with 0.1% v/v formic acid (mobile phase A) and 80% v/v acetonitrile with 0.1% v/v formic acid (mobile phase B), with peptides loaded and separated at a flow rate of 0.25 µl min⁻¹. A gradient was applied, starting with 2% to 9% B over 3 min, increasing to 30% B over 64 min, further to 44% B over 8 min and finally to 95% B over 5 min. Eluted peptides were ionized by an EASY-Spray source (Thermo Fisher Scientific) and introduced directly into the mass spectrometer, where MS data were acquired in data-dependent mode. The full scan mass spectrum was recorded with a resolution of 60,000. In each acquisition cycle, the ten most intense ions with a charge state from 2+ to 6+ were isolated and fragmented using higher-energy collisional dissociation with a normalized collision energy of 27%. The fragmentation spectra were recorded with a resolution of 50,000. Dynamic exclusion was enabled with a single repeat count and 30 s exclusion duration.

The remaining peptides were subjected to fractionation using size exclusion chromatography to enrich for cross-linked peptides[71]. Peptides were separated using a Superdex 30 Increase 3.2/300 column (GE Healthcare) at a flow rate of 10 µl min⁻¹. The earliest six peptide-containing fractions (fractions 6–11, 50 µl each) were collected, solvent removed using a vacuum concentrator, and then analyzed by LC–MS/MS. LC–MS/MS analysis of the size exclusion chromatography (SEC) fractions was performed using an Orbitrap Astral Mass Spectrometer (Thermo Fisher Scientific) connected to a Vanquish Neo UHPLC System (Thermo Fisher Scientific). Each SEC fraction was resuspended in a solution containing 1.6% v/v acetonitrile and 0.1% v/v formic acid. Peptides were injected onto a 5.5 cm High Throughput µPAC Neo HPLC Column (Thermo Scientific) operating at 50 °C column temperature. The mobile phase consisted of water

with 0.1% v/v formic acid (mobile phase A) and 80% v/v acetonitrile with 0.1% v/v formic acid (mobile phase B), with peptides separated at a flow rate of 0.3 µl min⁻¹. The separation gradient was optimized for each SEC fraction, ranging from 2% to 45–55% B over 88 min, followed by a ramp-up to 95% B within 7 min. Eluted peptides were ionized by an EASY-Spray source (Thermo Fisher Scientific) and introduced directly into the mass spectrometer.

For each SEC fraction, three LC–MS/MS analyses were conducted: the first without high field asymmetric waveform ion mobility spectrometry (FAIMS) applied, the second with FAIMS applied with a single compensation voltage (CV) filter and the third with FAIMS applied with two interval CV filters. The CV voltage applied for each SEC fraction was determined on the basis of previous optimization for complex peptide mixtures[72].

The MS data were acquired in the data-dependent mode with the top-speed option. For each 2.5 s acquisition cycle, the full scan mass spectrum was recorded in the Orbitrap with a resolution of 120,000. The ions with a charge state from 3+ to 7+ were isolated and fragmented using higher-energy collisional dissociation with a normalized collision energy of 30%. The fragmentation spectra were then recorded in the Astral mass analyzer. Dynamic exclusion was enabled with single repeat count and 60 s exclusion duration.

Protein identification was carried out using MaxQuant (version 2.4.10.0), searching against the *S. cerevisiae* protein sequence database downloaded from UniProt, with default search parameters and intensity-based absolute quantification (iBAQ) enabled.

For SEC fractions, MS2 peak lists were generated using the MSConvert module in ProteoWizard (version 3.0.11729), and identification of cross-linked peptides was performed using xiSEARCH software (version 1.7.6.4) against the sequences and reversed sequences of 315 proteins, including SPB subunits and co-isolated proteins identified with an intensity-based absolute quantification over $1 \times 10^6$ in the protein ID sample. The following parameters were applied for the search:

> MS accuracy = 3 ppm; MS2 accuracy = 5 ppm
> enzyme = trypsin (with full tryptic specificity);
> allowed number of missed cleavages = 3
> missing monoisotopic peak = 2
> cross-linker = BS3 (the reaction specificity for BS3 was assumed to be for lysine, serine, threonine, tyrosine and protein N termini)
> fixed modifications = carbamidomethylation on cysteine
> variable modifications = oxidation on methionine
> BS3 loop link
> hydrolyzed BS3 on one end
> Tris-reacted BS3 on one end
> maximum variable modification per peptide = 2

Cross-linked peptide candidates with a minimum of three matched fragment ions (with at least two containing a cross-linked residue) in each cross-linked peptide were filtered using xiFDR[73]. A false discovery rate of 10% at the residue-pair level was applied, with the 'boost between' option enabled for heteromeric cross-links.

MS data have been deposited in jPOST[74].

## Proteomics analysis of enriched SPBs

Twenty gel bands were cut out of a sodium dodecyl sulfate gel of the enriched SPBs. These gel samples were destained with 50% v/v acetonitrile and 50 mM ammonium bicarbonate, reduced with 10 mM DTT and alkylated with 55 mM iodoacetamide. Digestion was with 6 ng µl⁻¹ trypsin (Promega) overnight at 37 °C, and peptides were extracted in 2% v/v formic acid 2% v/v acetonitrile and analyzed by nano-scale capillary LC–MS/MS using an Ultimate U3000 HPLC (Thermo Fisher Scientific Dionex) to deliver a flow of approximately 300 nl min⁻¹. A µ-precolumn cartridge C18 Acclaim PepMap 100 (5 µm, 300 µm × 5 mm (Thermo Fisher Scientific Dionex) trapped the peptides before separation on a C18 Acclaim PepMap100 3 µm, 75 µm × 250 mm

(Thermo Fisher Scientific Dionex). Peptides were eluted with a 70 min gradient of acetonitrile (5% to 40%). The analytical column outlet was directly interfaced via a modified nano-flow electrospray ionization source, with a hybrid linear quadrupole ion trap mass spectrometer (Orbitrap QExactive, ThermoScientific). Data-dependent analysis was carried out, using a resolution of 60,000 for the full MS spectrum, followed by ten MS/MS spectra in the linear ion trap. MS spectra were collected over a $m/z$ range of 200–1,800. MS/MS scans were collected using threshold energy of 35 for collision-induced dissociation. LC–MS/MS data were then searched against the UNIPROT database, using the Mascot search engine program (Matrix Science). Database search parameters were set with a precursor tolerance of 10 ppm and a fragment ion mass tolerance of 0.8 Da. One missed enzyme cleavage was allowed, and variable modifications for oxidized methionine, carbamidomethyl and phosphorylated serine, threonine and tyrosine were included.

The MS proteomics data have been deposited to the ProteomeXchange Consortium via the PRIDE[75] partner repository with the dataset identifier PXD050372.

### Data visualization

All data were visualized and figures were prepared using ChimeraX[76]. The video was prepared using ArtiaX[77]. CLMS data were visualized and the figure was prepared using xiVIEW[78,79].

### Reporting summary

Further information on research design is available in the Nature Portfolio Reporting Summary linked to this article.

## Data availability

All data are available in the main text, Methods, Supplementary information and Extended data. Atomic coordinates and cryo-EM density maps of the γTuRC capping MTs (PDBs: 8QV2, 8QV3; maps: EMD-18665, EMD-18666) and MT lattice (PDB: 8QV0; map: EMD-18664) have been deposited in the Protein Data Bank (www.rcsb.org) and the Electron Microscopy Data Bank (https://ebi.ac.uk/pdbe/emdb/), respectively, and are also listed in Table 1. The atomic coordinates and crystallographic data for Spc98[NHD] have been deposited in the Protein Data Bank (www.rcsb.org) (accession code 8QRY) and are listed in Table 2. CLMS data have been deposited in jPOST (https://repository.jpostdb.org) accession code JPST002974 (PRIDE (https://www.ebi.ac.uk/pride) dataset identifier PXD050440). The MS proteomics data have been deposited to the ProteomeXchange Consortium via the PRIDE partner repository with the dataset identifier PXD050372. Source data are provided with this paper.

## Code availability

Source code for the Napari subboxer is available from GitHub at https://github.com/alisterburt/napari-subboxer.git (ref. 67).

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

## Acknowledgements

We are grateful to the LMB EM Facility for help with the EM data collection, J. Grimmett and T. Darling for computing, and T. Morgan and F. Begum of the Mass Spec Facility for proteomic analysis. We also thank members of Barford group for their input and useful discussions. J.V.K. is an Emeritus Staff Member of the MRC-LMB and is very grateful to S. Munro for providing a bench and facilities in his group. For the purpose of open access, the author has applied a CC BY public copyright license to any author accepted manuscript version arising. Grant funding was provided by UKRI/Medical Research Council MC_UP_1201/6 (Da.B.), Cancer Research UK C576/A14109 (Da.B.), Boehringer Ingleheim Fonds Fellowship (S.Y.) and Wellcome Trust [227434] (J.R.). The Wellcome Centre for Cell Biology is supported by core funding from the Wellcome Trust [203149] (J.R.). Deutsche Forschungsgemeinschaft (DFG, German Research Foundation) under Germany's Excellence Strategy – EXC 2008 – 390540038 – UniSysCat. Si-M/'Der Simulierte Mensch', a Science Framework of Technische Universität Berlin and Charité – Universitätsmedizin Berlin, Berlin, Germany, provided access to the mass spectrometer system (J.R.).

## Author contributions

Da.B., S.Y., T.D. and J.V.K. conceived the study. J.V.K., T.D. and S.Y. enriched the SPBs. T.D. and S.Y. prepared EM grids, collected and processed EM data, with help from A.B., and built the models. S.Y. purified and crystallized Spc98[NHD], and Do.B. collected and processed crystallography data. J.V.K. performed yeast genetic experiments. T.D. and J.V.K. cross-linked the enriched SPBs and Z.A.C. and J.R. performed the CLMS experiments on SPBs. S.Y., T.D., Da.B., J.V.K., Z.A.C. and J.R. analyzed the data and prepared the manuscript with input from all authors.

## Competing interests

The authors declare no competing interests.

## Additional information

**Extended data** is available for this paper at https://doi.org/10.1038/s41594-024-01281-y.

**Correspondence and requests for materials** should be addressed to Tom Dendooven, Stanislau Yatskevich, John V. Kilmartin or David Barford.

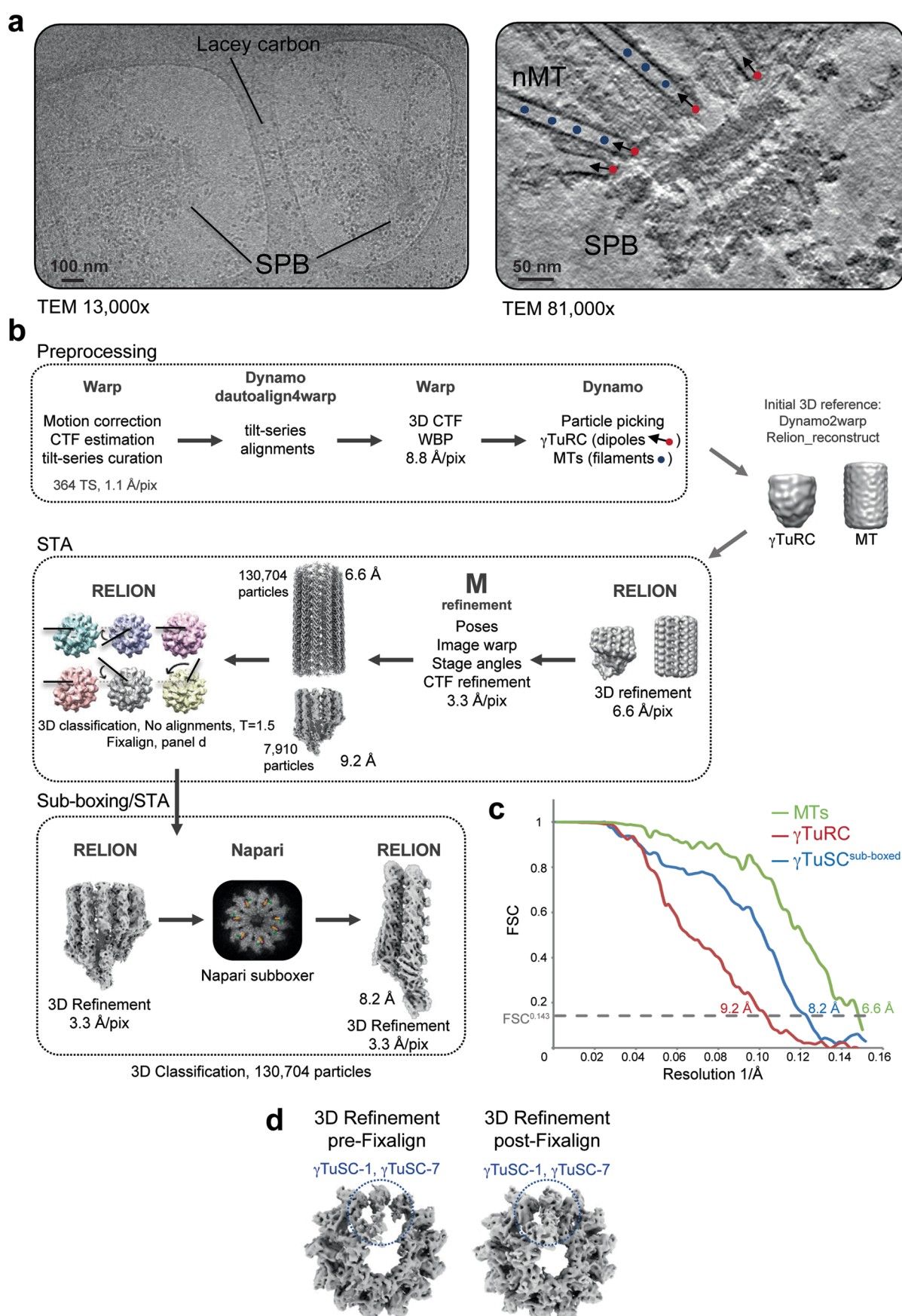

**Extended Data Fig. 1 | See next page for caption.**

**Extended Data Fig. 1 | Cryo-ET data processing scheme. a**, (left) Representative electron micrograph at 13,000x magnification of the enriched SPB sample on lacey grids. Similar SPB structures were observed in more than 500 independent tomograms. (right) Slice through a tomogram with red circles indicating centre of the dipole geometry used to pick γTuRCs, the arrows define the z-axis. Blue circles indicate positions along which microtubules were picked as a filament, the z-axis is pointed away from the SPB. Representative of more than 500 tomograms. **b**, Custom data processing pipeline, also outlined in the Methods section. Tilt-series (TS) were preprocessed in Warp and aligned using dautoalign4warp (Dynamo). Bin8 tomograms were generated in Warp and particle picking was performed in Dynamo, with γTuRC picked as dipoles and MTs picked as filaments. Sub-tomogram volumes were extracted in Warp and refined using RELION. Both γTuRC and MT particles were co-refined in M to give a final consensus refinement map. To identify and correct particle misalignments, a manual procedure (the Fixalign procedure) was developed: γTuRC particles were classified without alignment to identify classes where the seam of the γTuRC 3D class (full black line) was misaligned with respect to the seam of the initial γTuRC reference (dotted line). Misalignment was corrected by manually realigning 3D class averages in ChimeraX and applying the corresponding transformations to the respective particle stacks of the misaligned 3D class averages (arrow from full black line to dotted line). The Fixalign procedure improved density of γTuSCs-1 and γTuSCs-7 substantially (see also panel d). Next, A custom-built Napari sub-boxer plugin was used to sub-box and mask γTuRC particles into individual γTuSCs with two rows of α/β-tubulin to expand the dataset and improve the resolution. Sub-boxed γTuSCs were subjected to 3D refinement in RELION 3.1. WBP: weighted back projection. **c**, Fourier-shell correlation (FSC) curves for the reconstructions used in this study. **d**, Bottom view of the γTuRC before and after manually correcting misalignment of the γTuRC helical register (Fixalign). Prior to manual realignment the density for γTuSC-1 and γTuSC-7 was diffuse, but improved substantially after realignment and further 3D refinements.

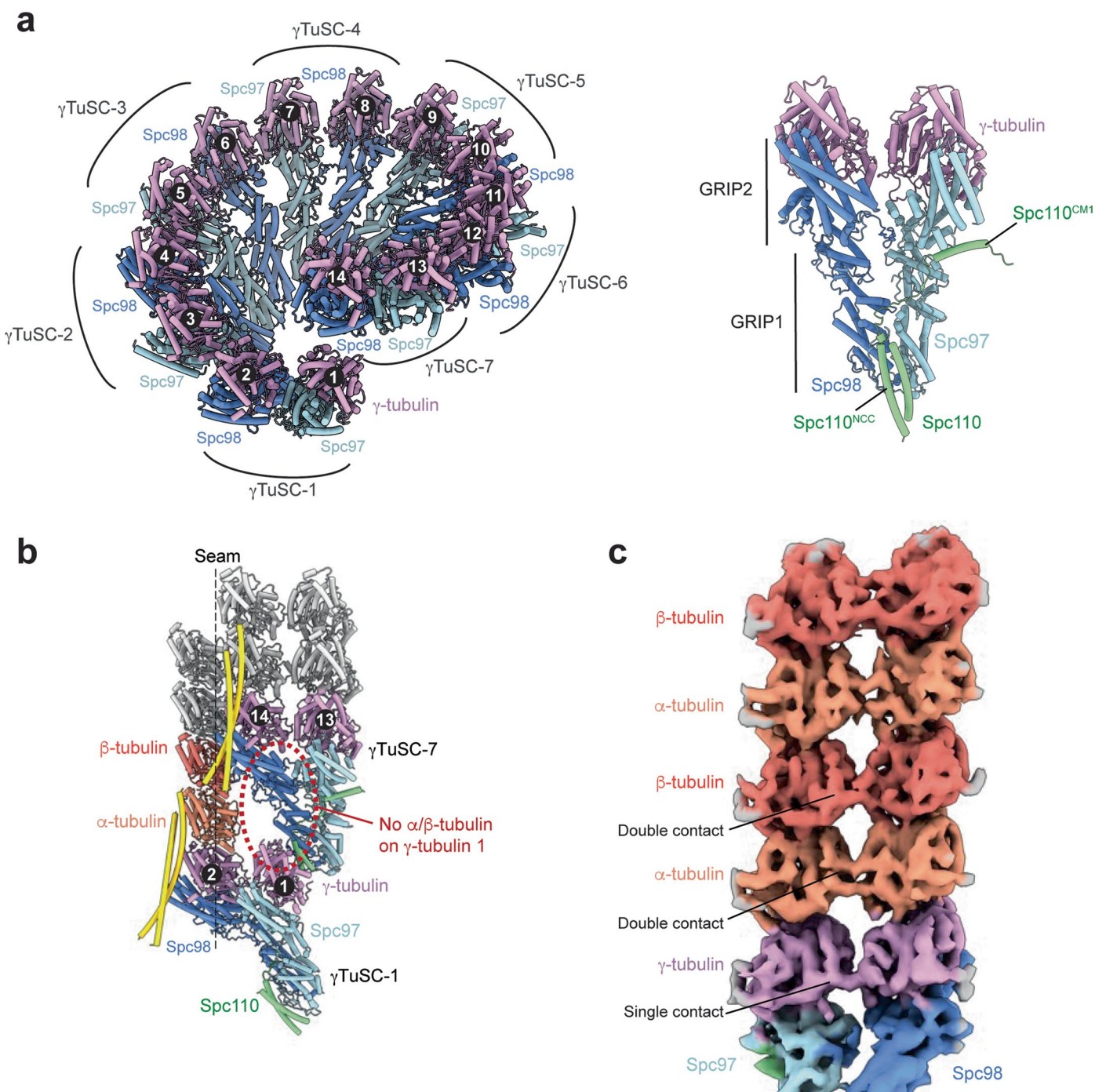

**Extended Data Fig. 2 | γTuRC geometry and interactions during MT nucleation. a**, Top-down view of γTuRC models with numbered γTuSCs and γ-tubulins (left). A molecular model of the γTuSC module with bound Spc110 protein and annotated GRIP1 and GRIP2 domains, is depicted on the right. **b**, Molecular model of the γTuRC seam shows that the first γ-tubulin bound to Spc97 is positioned precisely below the last Spc98 bound to the last (14th) γ-tubulin, with a large gap between them. This blocks α/β-tubulin binding to spoke 1 of γTuRC. **c**, Cryo-ET map of a sub-boxed γTuSC bound to the first two rows of α/β-tubulin, displaying the lateral α:α-tubulin (double contact), β:β-tubulin (double contact) and γ:γ-tubulin interfaces (single contact).

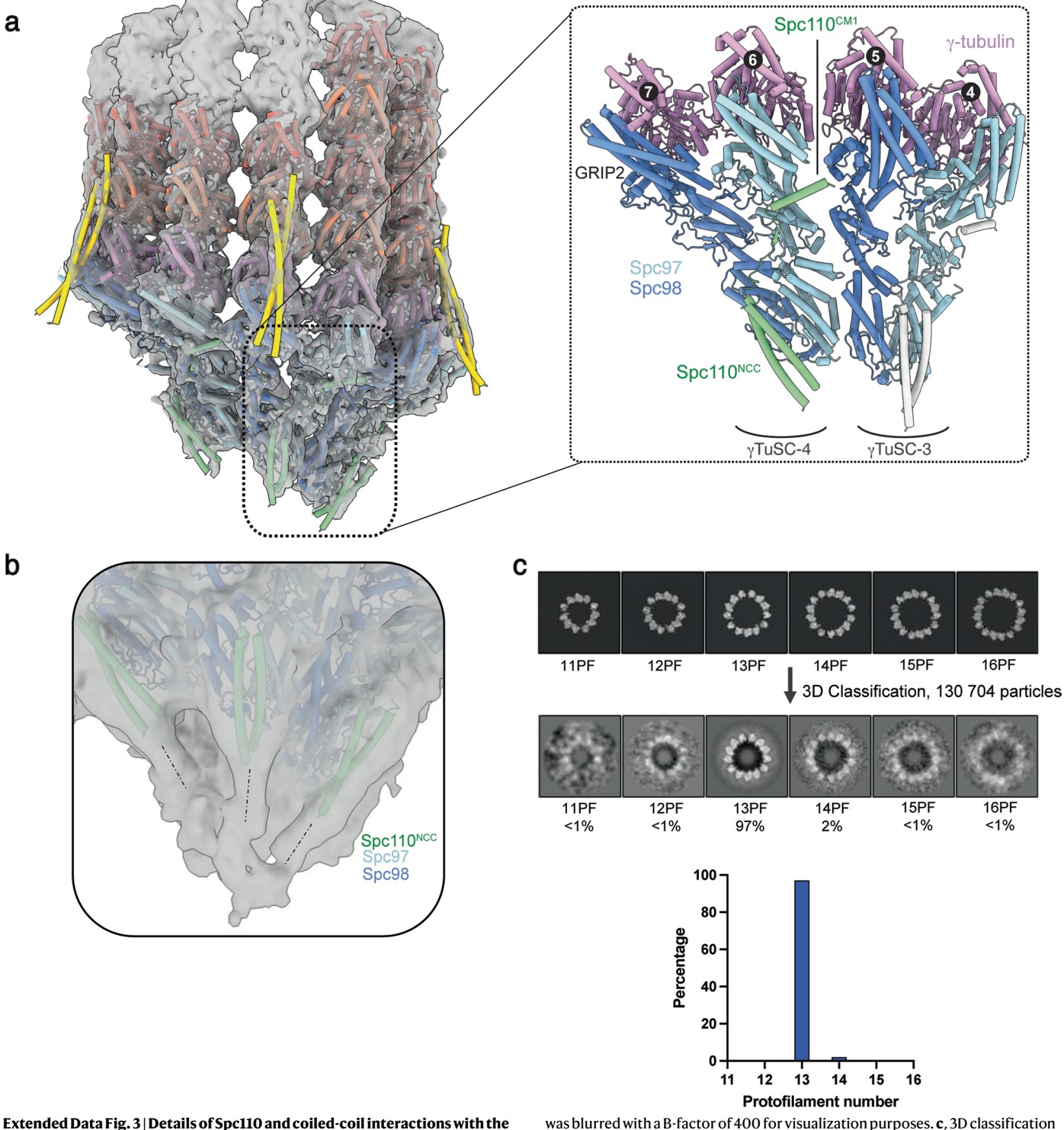

**Extended Data Fig. 3 | Details of Spc110 and coiled-coil interactions with the γTuRC. a**, Consensus cryo-ET map of the γTuRC with a fitted molecular model of the γTuRC shows that the secondary structure and path of Spc110[NCC] and Spc110[CMI] are resolved and in agreement with previous reconstituted structures[18]. **b**, At lower threshold the sub-tomogram average shows that Spc110[NCC] densities continue downwards until they converge on a single large stalk density. The map was blurred with a B-factor of 400 for visualization purposes. **c**, 3D classification of MTs was used to separate MT species according to protofilament numbers, following the MiRP procedure[68]. Representative slices through reference volumes used during 3D classification of the microtubule sub-tomogram volumes. 97% of spindle microtubules contain 13 protofilaments.

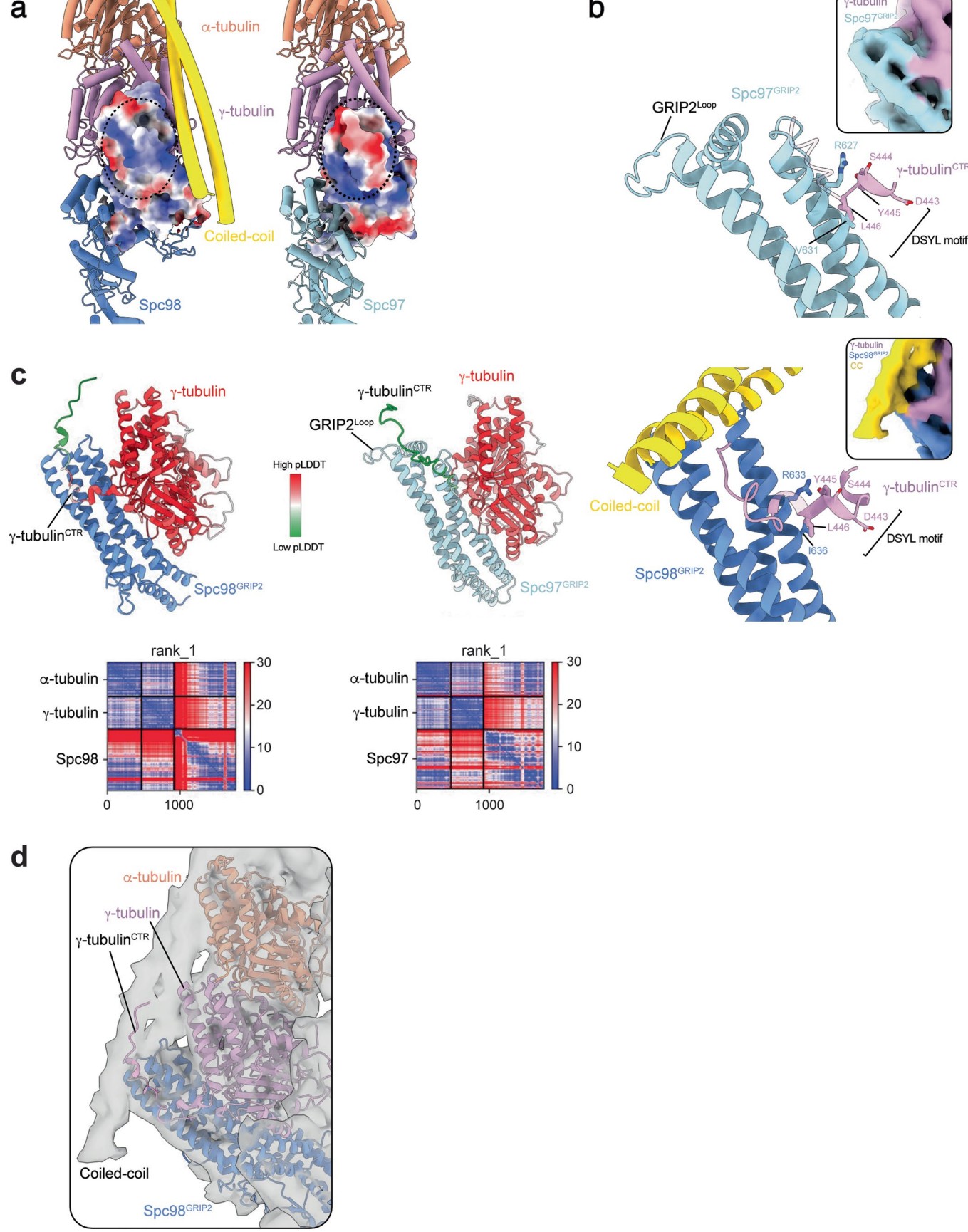

**Extended Data Fig. 4 | See next page for caption.**

**Extended Data Fig. 4 | Spc98 presents a unique interface to bind the coiled-coil density. a**, Electrostatic surface potential of Spc98 and Spc97 in the γTuSC structure shows that the coiled-coil protein is lined against a positively charged patch on Spc98$^{GRIP2}$. **b**, Alphafold2 predictions and cryo-EM density insets of the interface between the γ-tubulin$^{CTR}$ and Spc98$^{GRIP2}$ (top panel) and Spc97$^{GRIP2}$ (lower panel). The γ-tubulin$^{CTR}$ is predicted to bind extensively along the Spc98$^{GRIP2}$ domain and extends beyond the conserved 'DSYL' motif towards the coiled-coil density. At the Spc97$^{GRIP2}$ domain on the other hand, the γ-tubulin$^{CTR}$ tail is not predicted to interact beyond the 'DSYL' motif. **c**, γ-tubulin pLDDT scores mapped on the Alphafold2 predictions shown in panel b. γ-tubulin is coloured by the pLDDT score while Spc98 and Spc97 are coloured by subunit. PAE plots for the best model are also shown as insets. **d**, The unstructured γ-tubulin$^{CTR}$ extends toward the coiled-coil density in the γTuRC average.

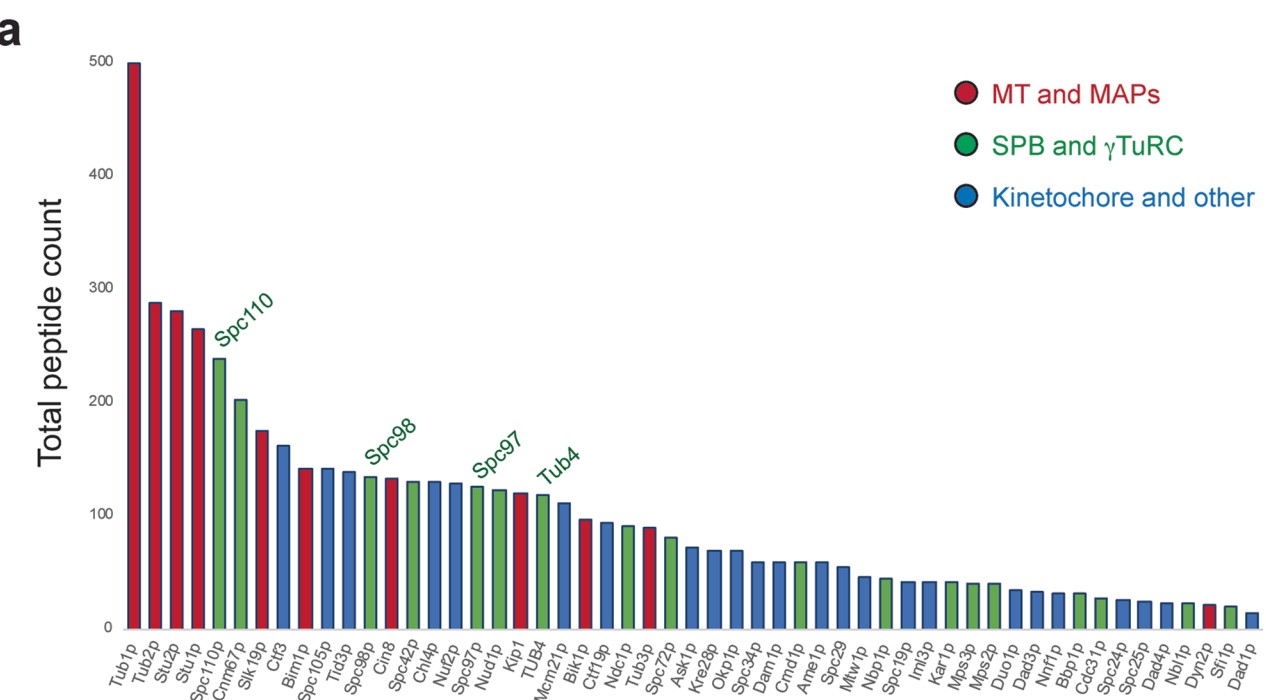

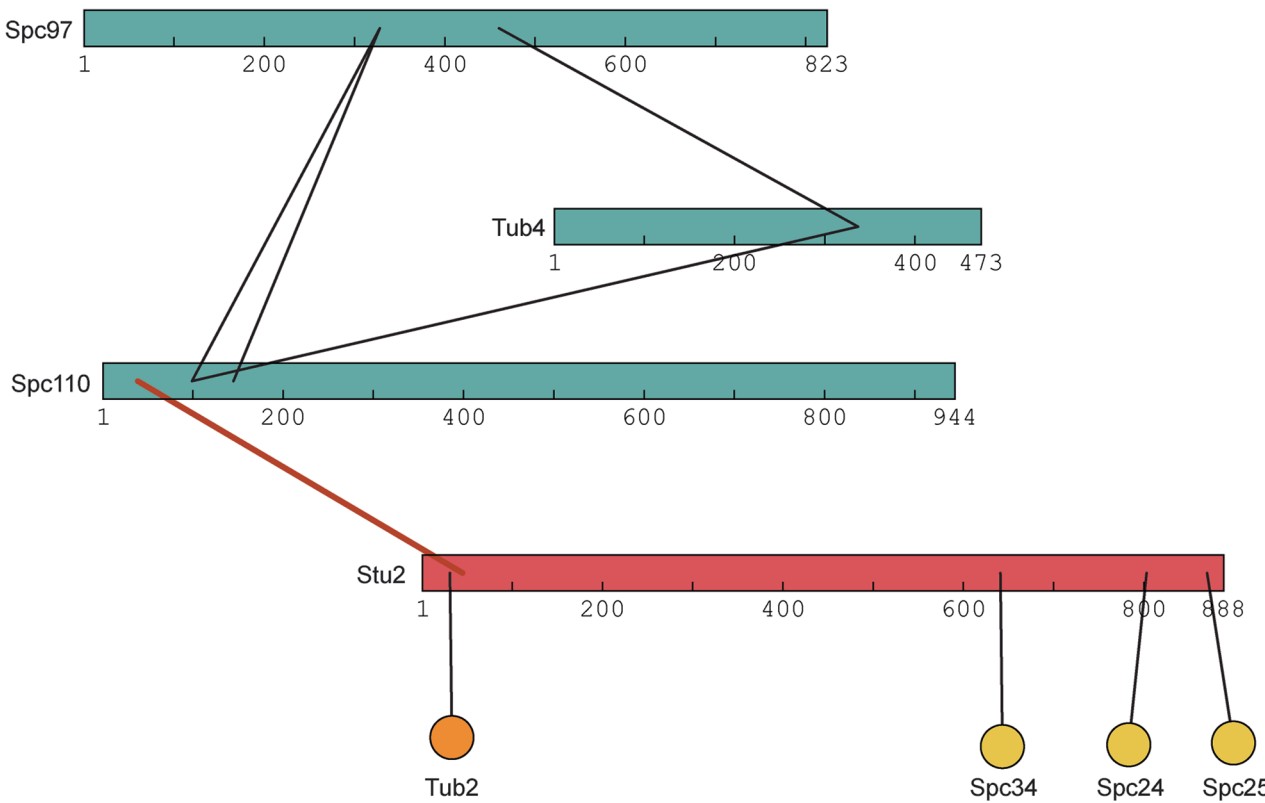

**Extended Data Fig. 5 | Mass spectrometry analysis of the components present in the SPB enrichment sample. a**, Mass spectrometry analysis of the enriched SPB sample, displaying only identified proteins with known function in microtubule–related processes. The full list of identified proteins is provided in the Supplementary Data 1 spreadsheet. **b**, Map of intermolecular cross-links between Stu2 and γTuRC components obtained through CL-MS analysis of the enriched SPB preparation. The Stu2 N-terminus crosslinks to the Spc110 N-terminus at γTuRC. Other proteins that were crosslinked to STU2 were shown as nodes.

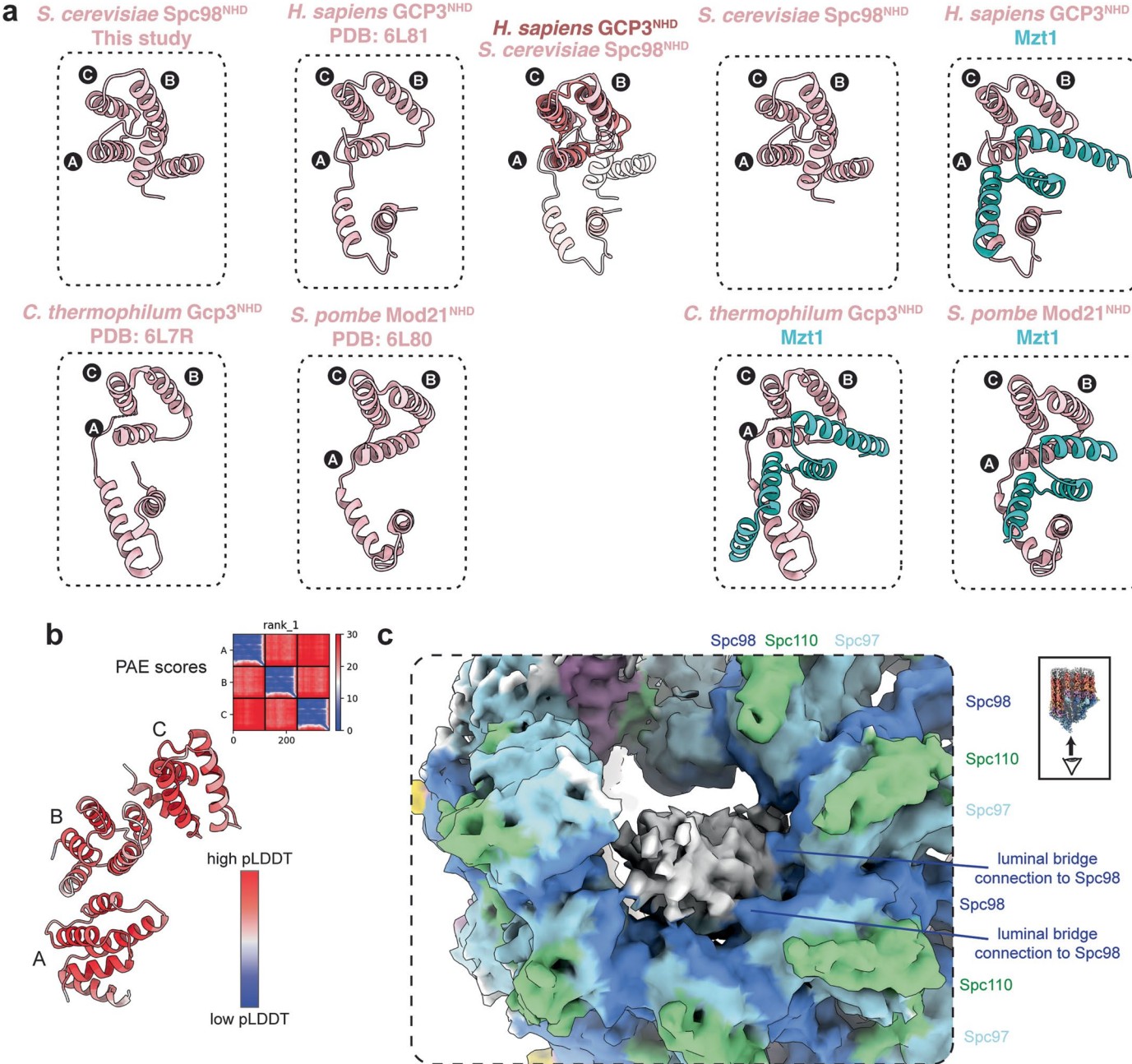

**Extended Data Fig. 6 | Structural comparison and analysis of the Spc98^NHD.**
**a**, Structures of the N-terminal Helical Domains (NHD) of Spc98 and homologs, without (left) and with (right) Mzt1 bound. The *S. cerevisiae* genome does not contain the *mzt1* gene. Structurally comparable helices are annotated with A, B and C to highlight similarities between Spc98^NHD and functionally equivalent proteins in other organisms. A direct overlay between the ABC helices of Spc98^NHD and GCP3^NHD is shown in the middle. GCP3^NHD and GCP6^NHD have very similar folds. **b**, Alphafold2 pLDDT PAE scores of the best trimeric Spc98^NHD prediction. **c**, Bottom view of the γTuRC map shows bridging connections between Spc98 modules and the luminal density.

# Reporting Summary

## Statistics

For all statistical analyses, confirm that the following items are present in the figure legend, table legend, main text, or Methods section.

| n/a | Confirmed | |
|---|---|---|
| ☒ | ☐ | The exact sample size (*n*) for each experimental group/condition, given as a discrete number and unit of measurement |
| ☒ | ☐ | A statement on whether measurements were taken from distinct samples or whether the same sample was measured repeatedly |
| ☒ | ☐ | The statistical test(s) used AND whether they are one- or two-sided<br>*Only common tests should be described solely by name; describe more complex techniques in the Methods section.* |
| ☒ | ☐ | A description of all covariates tested |
| ☒ | ☐ | A description of any assumptions or corrections, such as tests of normality and adjustment for multiple comparisons |
| ☒ | ☐ | A full description of the statistical parameters including central tendency (e.g. means) or other basic estimates (e.g. regression coefficient) AND variation (e.g. standard deviation) or associated estimates of uncertainty (e.g. confidence intervals) |
| ☒ | ☐ | For null hypothesis testing, the test statistic (e.g. $F$, $t$, $r$) with confidence intervals, effect sizes, degrees of freedom and $P$ value noted<br>*Give P values as exact values whenever suitable.* |
| ☒ | ☐ | For Bayesian analysis, information on the choice of priors and Markov chain Monte Carlo settings |
| ☒ | ☐ | For hierarchical and complex designs, identification of the appropriate level for tests and full reporting of outcomes |
| ☒ | ☐ | Estimates of effect sizes (e.g. Cohen's *d*, Pearson's *r*), indicating how they were calculated |

*Our web collection on statistics for biologists contains articles on many of the points above.*

## Software and code

Policy information about availability of computer code

| Data collection | EM data collection free software was used: SerialEM v4.2 for automated cryo-EM data collection |
|---|---|
| Data analysis | Cryo-electron tomography:<br>Micrograph movie frames were aligned with Warp v1.09. CTF estimation was performed using Warp v1.09. Tilts series alignment and particle picking were done using Dynamo v1.1.514. Sub-tomogram averaging was done in Relion v3.0 and v3.1 and M v1.09. Models were built using Chimera X v1.4 and COOT v0.8.9.2 and real-space refinement in PHENIX v1.18.2.<br><br>Model validation was performed using PHENIX v1.18.2. Visualization was done using Chimera X v1.4 and ArtiaX v0.3.<br>Structure prediction was performed with the ALPHAFOLD2 implemented in COLABFOLD v1.5.2.<br><br>Protein crystallography: Data were processed using MOSFLM v7.4.0 and AIMLESS v0.7.4 (CCP4 v8.0). The structure was determined using Phaser v2.8.3 and refined using Phenix v1.17.1.<br><br>Crosslinking mass spectrometry (CLMS): Protein identification was carried out using MaxQuant (v2.4.10.0).<br>For size exclusion chromatography fractions, MS2 peak lists were generated using the MSConvert module in ProteoWizard (v3.0.11729), and identification of crosslinked peptides was performed using xiSEARCH software (v1.7.6.4), and crosslinks were visualised in xiView v1.0.0.<br><br>The Napari sub-boxer code is available from https://github.com/alisterburt/napari-subboxer.git. |

For manuscripts utilizing custom algorithms or software that are central to the research but not yet described in published literature, software must be made available to editors and reviewers. We strongly encourage code deposition in a community repository (e.g. GitHub). See the Nature Portfolio guidelines for submitting code & software for further information.

## Data

Policy information about <u>availability of data</u>

All manuscripts must include a <u>data availability statement</u>. This statement should provide the following information, where applicable:

- Accession codes, unique identifiers, or web links for publicly available datasets
- A description of any restrictions on data availability
- For clinical datasets or third party data, please ensure that the statement adheres to our <u>policy</u>

Atomic coordinates and cryo-EM density maps of the γTuRC capping MTs (PDBs: 8QV2, 8QV3; maps: EMD-18665, EMD-18666), and MT lattice (PDB: 8QV0 ;map: EMD-18664) have been deposited in the Protein Data Bank (www.rcsb.org) and the Electron Microscopy Data Bank (https://ebi.ac.uk/pdbe/emdb/), respectively and also listed in Table 1.

The atomic coordinates and crystallographic data for Spc98NHD (PDB: 8QRY) have been deposited in the Protein Data Bank (www.rcsb.org) and listed in Table 2.

CLMS data have been deposited in jPOST (accession code JPST002974 [PRIDE dataset identifier PXD050440]).

The mass spectrometry proteomics data have been deposited to the ProteomeXchange Consortium via the PRIDE partner repository with the dataset identifier PXD050372.

Data availability.
All data are available in the main text, Methods and Supplementary and Extended Data. Atomic coordinates and cryo-EM density maps of the ▢TuRC capping MTs (PDBs: 8QV2, 8QV3; maps: EMD-18665, EMD-18666), and MT lattice (PDB: 8QV0 ;map: EMD-18664) have been deposited in the Protein Data Bank (www.rcsb.org) and the Electron Microscopy Data Bank (https://ebi.ac.uk/pdbe/emdb/), respectively and also listed in Table 1. The atomic coordinates and crystallographic data for Spc98NHD (PDB: 8QRY) have been deposited in the Protein Data Bank (www.rcsb.org) and listed in Table 2. CLMS data have been deposited in jPOST (accession code JPST002974 [PRIDE dataset identifier PXD050440]). The mass spectrometry proteomics data have been deposited to the ProteomeXchange Consortium via the PRIDE partner repository with the dataset identifier PXD050372.

Code availability.
Source code for the Napari sub-boxer is available from (https://github.com/alisterburt/napari-subboxer.git).

## Research involving human participants, their data, or biological material

Policy information about studies with <u>human participants or human data</u>. See also policy information about <u>sex, gender (identity/presentation), and sexual orientation</u> and <u>race, ethnicity and racism</u>.

| | |
|---|---|
| Reporting on sex and gender | N/A |
| Reporting on race, ethnicity, or other socially relevant groupings | N/A |
| Population characteristics | N/A |
| Recruitment | N/A |
| Ethics oversight | N/A |

Note that full information on the approval of the study protocol must also be provided in the manuscript.

# Field-specific reporting

Please select the one below that is the best fit for your research. If you are not sure, read the appropriate sections before making your selection.

☒ Life sciences          ☐ Behavioural & social sciences          ☐ Ecological, evolutionary & environmental sciences

For a reference copy of the document with all sections, see <u>nature.com/documents/nr-reporting-summary-flat.pdf</u>

# Life sciences study design

All studies must disclose on these points even when the disclosure is negative.

| | |
|---|---|
| Sample size | For the budding yeast spindle pole body, a total of 364 tilt series were collected. A total number of 7,910 particles were picked manually for yTuRC and used for the final reconstruction.  These are typical image numbers for cryo-ET data sets and were sufficient to obtain sub-nanometer reconstructions of yTuRC |
| Data exclusions | 3D classification procedures did not reveal broken particles or particles that do not belong to classes of interest, as all particles were picked manually for each tomogram. This is a standard practice in cryo-ET studies. |

| Replication | Cryo-ET datasets were collected with multiple samples in separate imaging sessions, all data collections were successful. All genetic experiments were repeated at least in three independent experiments and are all reproducible. |
|---|---|
| Randomization | For cryo-ET, sub-tomogram averages were randomized between even/odd groups during refinement and resolution estimation (gold-standard FSC). Randomisation is not applicable for the budding yeast mutants generated in this study which were assayed against the WT strain. |
| Blinding | For cryo-ET, all data collection and image processing procedures were automatically performed in an unbiased manner which is generally used in the field of cryo-EM and cryo-ET. |

# Reporting for specific materials, systems and methods

We require information from authors about some types of materials, experimental systems and methods used in many studies. Here, indicate whether each material, system or method listed is relevant to your study. If you are not sure if a list item applies to your research, read the appropriate section before selecting a response.

## Materials & experimental systems

| n/a | Involved in the study |
|---|---|
| ☒ ☐ | Antibodies |
| ☐ ☒ | Eukaryotic cell lines |
| ☒ ☐ | Palaeontology and archaeology |
| ☒ ☐ | Animals and other organisms |
| ☒ ☐ | Clinical data |
| ☒ ☐ | Dual use research of concern |
| ☒ ☐ | Plants |

## Methods

| n/a | Involved in the study |
|---|---|
| ☒ ☐ | ChIP-seq |
| ☒ ☐ | Flow cytometry |
| ☒ ☐ | MRI-based neuroimaging |

## Eukaryotic cell lines

Policy information about cell lines and Sex and Gender in Research

| Cell line source(s) | S. cerevisiae strain NCYC74 |
|---|---|
| Authentication | The S. cerevisiae strain was not authenticated |
| Mycoplasma contamination | The S. cerevisiae strain was not tested for mycoplasma contamination. |
| Commonly misidentified lines (See ICLAC register) | None |

## Plants

| Seed stocks | *Report on the source of all seed stocks or other plant material used. If applicable, state the seed stock centre and catalogue number. If plant specimens were collected from the field, describe the collection location, date and sampling procedures.* |
|---|---|
| Novel plant genotypes | *Describe the methods by which all novel plant genotypes were produced. This includes those generated by transgenic approaches, gene editing, chemical/radiation-based mutagenesis and hybridization. For transgenic lines, describe the transformation method, the number of independent lines analyzed and the generation upon which experiments were performed. For gene-edited lines, describe the editor used, the endogenous sequence targeted for editing, the targeting guide RNA sequence (if applicable) and how the editor was applied.* |
| Authentication | *Describe any authentication procedures for each seed stock used or novel genotype generated. Describe any experiments used to assess the effect of a mutation and, where applicable, how potential secondary effects (e.g. second site T-DNA insertions, mosiacism, off-target gene editing) were examined.* |

