## [Peer Review File · Nature Structural & Molecular Biology]

Peer Review Information

Manuscript Title: Structure of the native γ -Tubulin Ring Complex capping spindle microtubules

Corresponding author name(s): David Barford, Tom Dendooven, Stanislaw Yatskevich

Reviewer Comments & Decisions:

Decision Letter, initial version:

Message: 13th Dec 2023

Dear Dr. Barford,

Thank you again for submitting your manuscript "Structure of the native γ -Tubulin Ring Complex capping spindle microtubules". We now have comments (below) from the 3 reviewers who evaluated your paper. In light of those reports, we remain interested in your study and would like to see your response to the comments of the referees, in the form of a revised manuscript.

You will see that while reviewers appreciate the results, they raise several concerns which will need to be addressed in a revision. Specifically, reviewer #1 asks that you reconsider making conclusions regarding side chain positions in absence of high-resolution information and/or mutagenesis. Both reviewers #2 and #3 comment on the speculative nature of luminal density assignment – please consider their points to strengthen this conclusion or remove statements which are not experimentally validated. We agree with reviewer #3 that the coiled-coils' identity not being determined is a weakness of the manuscript, and would urge you to consider implementing cross-linking MS in revision.

Please be sure to address/respond to all concerns of the referees in full in a point-by-point response and highlight all changes in the revised manuscript text file. If you have comments that are intended for editors only, please include those in a separate cover letter.

We are committed to providing a fair and constructive peer-review process. Do not hesitate to contact us if there are specific requests from the reviewers that you believe are technically impossible or unlikely to yield a meaningful outcome. Please also remember that we are always open to scheduling a call if you would like to discuss a revision plan with us, or have any concerns.

We expect to see your revised manuscript within 5 months. If you cannot send it within this time, please contact us to discuss an extension; we would still consider your revision, provided that no similar work has been accepted for publication at NSMB or published elsewhere.

Reporting Summary:

When submitting the revised version of your manuscript, please pay close attention to our [href="https://www.nature.com/nature-portfolio/editorial-policies/image-integrity">Digital Image Integrity Guidelines](https://www.nature.com/nature-portfolio/editorial-policies/image-integrity). and to the following points below:

Please note that all key data shown in the main figures as cropped gels or blots should be presented in uncropped form, with molecular weight markers. These data can be aggregated into a single supplementary figure item. While these data can be displayed in a relatively informal style, they must refer back to the relevant figures. These data should be submitted with the final revision, as source data, prior to acceptance, but you may want to start putting it together at this point.

SOURCE DATA: we urge authors to provide, in tabular form, the data underlying the graphical representations used in figures. This is to further increase transparency in data reporting, as detailed in this editorial (<http://www.nature.com/nsmb/journal/v22/n10/full/nsmb.3110.html>). Spreadsheets can be submitted in excel format. Only one (1) file per figure is permitted; thus, for multi-paneled figures, the source data for each panel should be clearly labeled in the Excel file;

alternately the data can be provided as multiple, clearly labeled sheets in an Excel file. When submitting files, the title field should indicate which figure the source data pertains to. We encourage our authors to provide source data at the revision stage, so that they are part of the peer-review process.

Data availability: this journal strongly supports public availability of data. All data used in accepted papers should be available via a public data repository, or alternatively, as Supplementary Information. If data can only be shared on request, please explain why in your Data Availability Statement, and also in the correspondence with your editor. Please note that for some data types, deposition in a public repository is mandatory - more information on our data deposition policies and available repositories can be found below: <https://www.nature.com/nature-research/editorial-policies/reporting-standards#availability-of-data>

[Redacted]

Sincerely,
Kat

Katarzyna Ciazynska, PhD
(she/her)
Associate Editor
Nature Structural & Molecular Biology
<https://orcid.org/0000-0002-9899-2428>

Referee expertise:

Referee #1: cell biology, microtubule nucleation

Referee #2: cryo-EM, microtubules

Referee #3: cryo-EM, cryo-ET, microtubules

Reviewers' Comments:

Reviewer #1:

Remarks to the Author:

In this paper, Dendooven and colleagues analysed purified SPBs using cryo-EM to provide a 7-9 Å resolution structure of native budding yeast microtubule-attached γ -TuRCs. Prior to this work, several relatively high-resolution structures, mainly recombinant, of budding yeast γ -TuRC and vertebrate γ -TuRC without an associated microtubule have been published. There was also a low-resolution structure of a native budding yeast γ -TuRC attached to a microtubule published in 2015, but this did not provide clear molecular detail. The advance here is providing a relatively high-resolution structure of a native γ -TuRC in a state where it has nucleated a microtubule i.e. in an active state, as opposed to the low activity states observed previously. This helps to address the next big question in the field, which is how are γ -TuRCs activated – how do they transition from an open low-activity to a closed active state. The authors convincingly show that the budding yeast γ -TuRC is a perfect template for a 13-protofilament microtubule. More importantly, the high-resolution structure enabled the authors to make 2 key observations: 1) budding yeast γ -TuRCs have a luminal bridge composed of a trimer of Spc98-NTD's (lacking Mzt1 and actin); 2) a coiled-coil domain binds across Spc98, γ -tubulin and the first row of α/β -tubulin dimers, appearing to either promote nucleation or stabilise the γ -TuRC-MT attachment, or both. These observations provide an important conceptual advance. The data is solid and I only have some relatively minor concerns that can be addressed textually.

Major comments.

1. In the abstract, I feel several text changes need to be made:

A) " β -tubulin capping the more dynamic MT 'plus' end". B-tubulin does not really "cap" the plus end, it is just exposed at the plus end.

B) "Because all known γ TuRC structures are devoid of MTs and exhibit an 'open', inactive conformation..." Firstly, this should be "low activity conformation" as these complexes are capable of nucleating microtubules. Moreover, previous structures of budding yeast γ -TuRC are in a near-closed conformation. So this is a statement that does not take into

account some important nuances (see also point 2 below).

C) "in the context of enriched yeast mitotic spindles". Need to make it clear which yeast as budding yeast and fission yeast have very different γ -TuRCs (additional proteins found in fission yeast). This point is relevant in other parts of the manuscript where the authors refer to yeast.

D) "In our structure, γ TuRC adopts an active closed conformation to function as a perfect geometric helical template presenting a ring of g-tubulin subunits to seed nucleation of exclusively 13-protofilament microtubules." This makes it sound as though the γ -TuRC adopts the perfect geometry before nucleation, but later in the paper the authors suggest that α/β -tubulin binding will help close the γ -TuRC. My take, from this current paper and from Brilot et al., 2021, is that Spc110 stimulates the assembly of γ -TuSCs into a ring structure that samples the closed conformation and that this conformation is locked in only once α/β -tubulin dimers are added and form lateral interactions. So I think the above statement needs rephrasing to reflect this.

E) "This positioning of α/β -tubulin onto γ TuRC reveals a role for the coiled-coil protein in augmenting γ TuRC-mediated microtubule nucleation." Without experimental evidence, I think the authors can say that this "suggests" a role, but not that it "reveals" a role.

2. In the introduction, the authors say "All available γ TuRC structures are devoid of MTs and exhibit an 'open', inactive conformation". Firstly, Kollman et al., 2015 show a 38 Angstrom structure of yeast γ -TuRC bound to microtubule. Secondly, the yeast γ -TuRC structure shown in Brilot et al., 2021 is very different from the structures of *Xenopus* and human γ -TuRCs. The yeast structure is in a conformation much closer to a closed "active" state and even samples the active state (which is why crosslinking experiments to close the γ -TuRC worked previously). Later on, the authors even say that "The closed geometry of native γ TuRC is similar but not identical to that of a recombinant γ TuSC filament trapped in a closed state by engineered disulphides in g-tubulin". So I think the statement in the introduction is a generalisation that does not take into account some details that would be beneficial to the reader. Most importantly, I think it is evident from the Brilot et al. data and from the data in this current paper that yeast γ -TuRCs are assembled into a near-closed state that sample the closed state and that then become locked in a closed state during nucleation (by lateral α/β -tubulin interactions). So the authors need to be careful how they introduce the prior observations of budding yeast γ -TuRC and make sure they are distinguished from vertebrate γ -TuRC.

3. End of introduction: "Our structure of a native γ TuRC capping a microtubule explains the molecular mechanism of γ TuRC-mediated MT nucleation". I think this is an overstatement. There is still more to learn about the molecular mechanism – for example, we don't yet even know the identity of the coiled-coil. Moreover, budding yeast γ -TuRCs are simple relative to those in other Eukaryotes and so there is even more to learn there. I think the authors should focus their statement on revealing the coiled-coil and the luminal bridge as these findings are truly novel.

4. The identification of a coiled-coil that may support nucleation and microtubule anchoring is very interesting and it is a shame that its identity cannot be revealed. The authors state that "Despite extensive biochemical experiments, we were unable to validate a candidate with high confidence". Can the authors please discuss what has been tried and how it failed (to avoid others wasting their time trying the same thing)?

5. The authors state that the luminal bridge in vertebrate γ -TuRCs is formed by "Mzt1 intertwined with either the GCP3 or GCP5 (Spc98 homologues) N-terminal domains

(GCP3N and GCP5N)", but I believe it is formed with the N-terminal domains of GCP3 and GCP6. This should be corrected.

6. When the authors compare the Mzt1/GCP-NTD modules with Spc98-NTD, they say "Interestingly, *S. cerevisiae* Spc98NHD is similar in size and fold to the composite Mzt1-GCP3N/GCP5N structures in higher eukaryotes, with the more compact Spc98NHD helical bundle accounting for both the Mzt1 and GCP3N/GCP5N helices at the luminal bridge (Extended Data Fig. 5a)." I do not really see a great similarity between Spc98-NHD and the MZT1/GCP modules. Budding yeast Spc98-NHD still has only 5 helical bundles, so does not directly account for the additional 3 Mzt1 helices. Also, the human GCP3-NHD is not shown in the figure. Can the authors better show the similarity, with an overlay perhaps? Are they simply saying that the Spc98-NHD is more compact and that is what makes up for the lack of MZT1?

7. In the conclusion: "We describe the first structure of a native γ TuRC capping nuclear microtubule (nMT) minus-ends at the MTOC, obtained through sub-tomogram averaging" needs to be toned down or re-worded as Kollman et al., 2015 showed a structure, albeit of lower resolution.

Reviewer #2:

Remarks to the Author:

The authors in this manuscript describe the structural organization of a gammaTURC complex using sub-tomogram averaging on isolated spindle pole bodies. The authors show that the gammaTURC complex nicely accommodates a 13-protofilament microtubule. Moreover, the authors discover a new structural feature of the microtubule gammaTURC ends with a never-before-seen coiled-coil protein. The quality of the 3D reconstruction and the data analysis is commendable and has clear figures and layouts.

Overall, the cryo-ET analysis is rigorous, general structure interpretation is grounded in data and figures are of high quality. My only comments are minor.

Luminal bridge interpretation: It is interesting to see the luminal bridge density inside of the ring, especially considering that yeast is different than the metazoan gammaTURCs. That said, docking an AlphaFold prediction into the density without further validation is highly speculative and requires validation. The authors need to validate this using an experimental approach or remove the docking. Moreover, the PAE plot in Extended Data Figure 5b does not show a very strong signal for the 3D organization of a trimer.

Structure interpretation and figures: The authors are over-interpreting their models, considering they do not have density or mutagenesis to validate their models when they show side chains. As such, showing side chains is much too speculative. In general, modeling inter-amino acid contacts without mutations or density is an over-interpretation. As such, I request that the authors roll back their claims regarding side-chain contacts. I ask that the authors remove side chain interactions if they do not mutate the residues or have density. Examples include Figure 2e, 3d, 3e. Extended Data Figure 2c.

Final model figure: The authors are proposing a specific model of how the coiled-coil protein facilitates nucleation without any data for this model. The authors do not know, for instance, if the coiled-coil protein binds simultaneously with a/b tubulin or subsequently

after the microtubule is formed. The overall model is exciting to consider, but these intermediate steps are not grounded in data. Finally, Figure 4d is similarly speculative and not ground in their structural data. The authors should only be showing models that is supported by experimental data.

Reviewer #3:

Remarks to the Author:

In this article Dendooven et al report the cryo-ET structure of the γ -Tubulin Ring Complex (γ TuRC) from yeast. γ TuRC were from enriched mitotic spindles. In their preparation, γ TuRC formed a closed conformation and included microtubules (MTs). The overall architecture of γ TuRC resolved here was similar to previously published ones, i.e. seven γ TuSC (made of γ tubulin, Spc97/98) forming a ring. The main difference to previous studies was the diameter of the ring. In addition, the resolved cryo-ET structure has extra densities, one corresponding to unidentified protein that seems to adopt a coiled-coil structure and binds to the first row of α/β -tubulin. The second density, an elongated luminal density that bridges γ TuSC. The authors suggest that this second density is three copies of the N-terminal helical domain of Spc98 (Spc98NHD), based on an AlphaFold trimer prediction based on their crystal structure of monomeric Spc98NHD. Finally, the authors propose a mechanism for γ TuRC assembly and MTs nucleation (Figure 4c and d).

The data presented and image processing is of high quality and I do not have any concern about the presented structure.

The weakness of this study is the limited resolution achieved ($\sim 8-9\text{\AA}$), which meant the 'coiled-coil' density remained unassigned and the assignment for the luminal density is missing further validation, especially that the resolution of that part of the map looks even worse.

Taken that the 'coiled-coil' density is the only major novel aspect of this structure, I am afraid the impact of this structure is very limited.

Assuming it is not possible to get higher resolution structure, the authors should do cross-linking mass spectrometry to identify this density.

Minor comments are related to the way the data is presented. The manuscript in its current form assumes previous knowledge on γ TuRC, there are many abbreviations of domains and sub-complexes that are not annotated in the main figures (only in the supplement). Below are specific suggestions to make it more accessible for the general reader:

- Spokes and γ TuSC should be annotated clearly in Figure 1. Incorporate Extended Data Fig. 2a and b in Figure 1.
- Figure 2b-d can be in the supplement and the font should be bigger for the annotations
- Figure 4d should be removed it is duplication of c(iii)
- Extended Data Fig. 6 should be a main figure and the model in b more visible

Line 257, the call for Extended Data Fig. 5b is not correct

Lastly, Extended Data Tables need consistency when using dots/commas/space with the numbers. Look for example on 'Initial/final particle images (no.)', also is it number of particles or number of 'images' (micrographs)?

Author Rebuttal to Initial comments

Referee expertise:

Referee #1: cell biology, microtubule nucleation

Referee #2: cryo-EM, microtubules

Referee #3: cryo-EM, cryo-ET, microtubules

Reviewers' Comments:

We thank all three referees for their reviews of the manuscript and for their constructive suggestions to improve this study and to clarify the text and figures. Reviewers' comments in black, our responses in red. Changes to the manuscript are also in red.

In the manuscript, we have discussed our study in context of the recent published paper from the Llorca and Surrey groups (PMID 38305685) and the posted preprint of Kapoor and colleagues (<https://doi.org/10.1101/2023.11.20.567916>). Additionally we include a Supplementary Video that shows the cryo-electron tomogram and annotates the positions of γ TuRC capped microtubules.

Reviewer #1:

Remarks to the Author:

In this paper, Dendooven and colleagues analysed purified SPBs using cryo-EM to provide a 7-9 Å resolution structure of native budding yeast microtubule-attached γ -TuRCs. Prior to this work, several relatively high-resolution structures, mainly recombinant, of budding yeast γ -TuRC and vertebrate γ -TuRC without an associated microtubule have been published. There was also a low-resolution structure of a native budding yeast γ -TuRC attached to a microtubule published in 2015, but this did not provide clear molecular detail. The advance here is providing a relatively high-resolution structure of a native γ -TuRC in a state where it has nucleated a microtubule i.e. in an active state, as opposed to the low activity states observed previously. This helps to address the next big question in the field, which is how are γ -TuRCs activated – how do they transition from an open low-activity to a closed active state. The authors convincingly show that the budding yeast γ -TuRC is a perfect template for a 13-protofilament microtubule. More importantly, the high-resolution structure enabled the authors to make 2 key observations: 1) budding yeast γ -TuRCs have a luminal bridge composed of a trimer of Spc98-

NTD's (lacking Mzt1 and actin); 2) a coiled-coil domain binds across Spc98, γ -tubulin and the first row of α/β -tubulin dimers, appearing to either promote nucleation or stabilise the γ -TuRC-MT attachment, or both. These observations provide an important conceptual advance.

The data is solid and I only have some relatively minor concerns that can be addressed textually.

We thank the reviewer for their assessment of our manuscript and for suggestions to improve the clarity of the manuscript and to make it more consistent with known and published results.

Major comments.

1. In the abstract, I feel several text changes need to be made:

A) " β -tubulin capping the more dynamic MT 'plus' end". β -tubulin does not really "cap" the plus end, it is just exposed at the plus end. B) "Because all known γ TuRC structures are devoid of MTs and exhibit an 'open', inactive conformation..." Firstly, this should be "low activity conformation" as these complexes are capable of nucleating microtubules. Moreover, previous structures of budding yeast γ -TuRC are in a near-closed conformation. So this is a statement that does not take into account some important nuances (see also point 2 below). C) "in the context of enriched yeast mitotic spindles". Need to make it clear which yeast as budding yeast and fission yeast have very different γ -TuRCs (additional proteins found in fission yeast). This point is relevant in other parts of the manuscript where the authors refer to yeast.

D) "In our structure, γ TuRC adopts an active closed conformation to function as a perfect geometric helical template presenting a ring of γ -tubulin subunits to seed nucleation of exclusively 13-protofilament microtubules." This makes it sound as though the γ -TuRC adopts the perfect geometry before nucleation, but later in the paper the authors suggest that α/β -tubulin binding will help close the γ -TuRC. My take, from this current paper and from Brilot et al., 2021, is that Spc110 stimulates the assembly of γ -TuSCs into a ring structure that samples the closed conformation and that this conformation is locked in only once α/β -tubulin dimers are added and form lateral interactions. So I think the above statement needs rephrasing to reflect this. E) "This positioning of α/β -tubulin onto γ TuRC reveals a role for the coiled-coil protein in augmenting γ TuRC-mediated microtubule nucleation." Without experimental evidence, I think the authors can say that this "suggests" a role, but not that it "reveals" a role.

We thank the reviewer for these helpful comments and have revised the Abstract accordingly.

2. In the introduction, the authors say “All available γ TuRC structures are devoid of MTs and exhibit an ‘open’, inactive conformation”. Firstly, Kollman et al., 2015 show a 38 Angstrom structure of yeast γ -TuRC bound to microtubule. Secondly, the yeast γ -TuRC structure shown in Brilot et al., 2021 is very different from the structures of *Xenopus* and human γ -TuRCs. The yeast structure is in a conformation much closer to a closed “active” state and even samples the active state (which is why crosslinking experiments to close the γ -TuRC worked previously). Later on, the authors even say that “The closed geometry of native γ TuRC is similar but not identical to that of a recombinant γ TuSC filament trapped in a closed state by engineered disulphides in g-tubulin”. So I think the statement in the introduction is a generalisation that does not take into account some details that would be beneficial to the reader. Most importantly, I think it is evident from the Brilot et al. data and from the data in this current paper that yeast γ -TuRCs are assembled into a near-closed state that sample the closed state and that then become locked in a closed state during nucleation (by lateral α/β -tubulin interactions). So the authors need to be careful how they introduce the prior observations of budding yeast γ -TuRC and make sure they are distinguished from vertebrate γ -TuRC.

We thank the reviewer for highlighting differences between available yeast and vertebrate γ TuRC structures. In the light of recently published work by the Llorca and Surrey’s labs, we have removed the statement “All available γ TuRC structures are devoid of MTs and exhibit an ‘open’, inactive conformation”. We follow it with more precise descriptions of vertebrate γ TuRCs and then separate mention of yeast γ TuRCs. (page 2 lines 3-6, lines 8-11).

3. End of introduction: “Our structure of a native gTuRC capping a microtubule explains the molecular mechanism of gTuRC-mediated MT nucleation”. I think this is an overstatement. There is still more to learn about the molecular mechanism – for example, we don’t yet even know the identity of the coiled-coil. Moreover, budding yeast γ -TuRCs are relatively simple compared to those in other Eukaryotes and so there is even more to learn there. I think the authors should focus their statement on revealing the coiled-coil and the luminal bridge as these findings are truly novel.

We agree that this statement is too strong and have changed 'explains' to 'expand our understanding of'. (page 2 line 32)

4. The identification of a coiled-coil that may support nucleation and microtubule anchoring is very interesting and it is a shame that its identity cannot be revealed. The authors state that "Despite extensive biochemical experiments, we were unable to validate a candidate with high confidence". Can the authors please discuss what has been tried and how it failed (to avoid others wasting their time trying the same thing)?

Stu2 is a strong candidate for the coiled-coil protein based on available literature. It was an abundant protein in our SPB preparation (Extended Data Fig. 5a), and has a coiled-coil structure that fits the coiled-coil density. However, size exclusion chromatography analysis of Spc97:Spc98: γ -tubulin:Spc110 (γ TuRC) with full-length Stu2 showed no binding of Stu2 to the assembled γ TuRC, as assessed by *in vitro* CLMS and SDS PAGE. We have discussed this in the text (page 5).

To further address the identity of the coiled-coil protein, we collaborated with the group of Juri Rappsilber to use cross-linking mass spectrometry (CLMS) of the enriched SPB preps. This revealed many cross-links that are consistent with the known interactions of γ TuRC, SPB plaques and kinetochore (Fig. 3d). Although CLMS did not detect any other proteins interacting directly with either Spc98 or γ -tubulin, three proteins were identified as having previously unknown cross-links to Spc110, two of which are coiled-coil proteins: Bre1 and Stu2. Only Stu2 is enriched at SPBs, crosslinking near the Spc110 N-terminus, but not directly via its coiled-coil (Fig. 3d and Extended Data Fig. 5a,b).

Although we have narrowed down the list of potential coiled-coil candidates, we were unable to unambiguously identify this protein. We discuss the implications in the text (page 5, from line 45).

5. The authors state that the luminal bridge in vertebrate γ -TuRCs is formed by "Mzt1 intertwined with either the GCP3 or GCP5 (Spc98 homologues) N-terminal domains (GCP3N and GCP5N)", but I believe it is formed with the N-terminal domains of GCP3 and GCP6. This should be corrected.

We thank the reviewer for spotting this mistake and have corrected it in the text (page 6 and through the text) and in the Extended Data Fig. 5.

6. When the authors compare the Mzt1/GCP-NTD modules with Spc98-NTD, they say “Interestingly, *S. cerevisiae* Spc98NHD is similar in size and fold to the composite Mzt1-GCP3N/GCP5N structures in higher eukaryotes, with the more compact Spc98NHD helical bundle accounting for both the Mzt1 and GCP3N/GCP5N helices at the luminal bridge (Extended Data Fig. 5a).” I do not really see a great similarity between Spc98-NHD and the MZT1/GCP modules. Budding yeast Spc98-NHD still has only 5 helical bundles, so does not directly account for the additional 3 Mzt1 helices. Also, the human GCP3-NHD is not shown in the figure. Can the authors better show the similarity, with an overlay perhaps?

We have adjusted the text to better convey the message that the Spc98^{NHD} is a compact structure that does not resemble very well the more extended GCP3/6^{NHD} folds alone, but bears more similarity to the composite one-half of the Mzt1-GCP3/6^{NHD} fold (five α -helices). We remade Extended Data Fig. 5a (now ED Fig. 6a) to better show the similarity in orientation and topology of helices A, B and C between Spc98^{NHD} and GCP3/6^{NHD}. The reviewer is also correct to succinctly state that Spc98-NHD is simply more compact by itself and this does make up for the lack of Mzt1 in budding yeast. We modified the main text to reflect this (page 6, lines 18-25). We have also included an extra inset showing the overlay between Spc98^{NHD} and GCP3^{NHD}. GCP3^{NHD} and GCP6^{NHD} are near identical, hence we only show GCP3^{NHD}.

7. In the conclusion: “We describe the first structure of a native γ TuRC capping nuclear microtubule (nMT) minus-ends at the MTOC, obtained through sub-tomogram averaging” needs to be toned down or re-worded as Kollman et al., 2015 showed a structure, albeit of lower resolution.

We have reworded this sentence: “We describe 7-9 Å resolution structure of a native γ TuRC capping nuclear microtubule (nMT) minus-ends at the MTOC, obtained through sub-tomogram averaging”. (page 6, line 49).

Reviewer #2:

Remarks to the Author: The authors in this manuscript describe the structural organization of a

gammaTURC complex using sub-tomogram averaging on isolated spindle pole bodies. The authors show that the gammaTURC complex nicely accommodates a 13-protofilament microtubule. Moreover, the authors discover a new structural feature of the microtubule gammaTURC ends with a never-before-seen coiled-coil protein. The quality of the 3D reconstruction and the data analysis is commendable and has clear figures and layouts. Overall, the cryo-ET analysis is rigorous, general structure interpretation is grounded in data and figures are of high quality. My only comments are minor.

We thank the reviewer for their assessment of our manuscript and helpful suggestions to improve it.

1. Luminal bridge interpretation: It is interesting to see the luminal bridge density inside of the ring, especially considering that yeast is different than the metazoan gammaTURCs. That said, docking an AlphaFold prediction into the density without further validation is highly speculative and requires validation. The authors need to validate this using an experimental approach or remove the docking. Moreover, the PAE plot in Extended Data Figure 5b does not show a very strong signal for the 3D organization of a trimer.

We agree with the referee that docking the Spc98^{NHD} into the cryo-ET density is too speculative without further validation. We performed CLMS on the enriched SPBs during the revision of this manuscript, in an attempt to find crosslinks between Spc98^{NHD} and Spc97/98 GRIP1 domains, but did not observe any crosslinks. We have therefore removed the docking of the Alphafold model in the density in Fig. 4b, and we toned down the language we use to assign the luminal bridge as Spc98^{NHD} trimer (page 6 lines 22-27). We agree that the PAE score for the higher order organisation of Spc98^{NHD} is not very strong, but in our view, it is still valuable to include as it points towards a tendency for Spc98^{NHD} to form multimers.

2. Structure interpretation and figures: The authors are over-interpreting their models, considering they do not have density or mutagenesis to validate their models when they show side chains. As such, showing side chains is much too speculative. In general, modelling inter-amino acid contacts without mutations or density is an over-interpretation. As such, I request that the authors roll back their claims regarding side-chain contacts. I ask that the authors remove side chain interactions if they do not mutate the residues or have density. Examples include Figure 2e, 3d, 3e. Extended Data Figure 2c.

We agree with the referee that we should be more careful with showing side chains. We have therefore:

- Replaced insets i and ii in Fig 2e with electrostatic surfaces to show that the interactions are electrostatic without assuming side chain geometries.
- Removed insets with side chains in Extended Data Fig. 2c.
- Removed most side chains in Fig. 3c, leaving only the ones for which we have biochemical data (Fig. 3e).
- Removed most side chains in Fig. 3d, leaving only the ones for which there is support from high resolution structures of γ TuSC, now cited in the main text as well.

3. Final model figure: The authors are proposing a specific model of how the coiled-coil protein facilitates nucleation without any data for this model. The authors do not know, for instance, if the coiled-coil protein binds simultaneously with α/β tubulin or subsequently after the microtubule is formed. The overall model is exciting to consider, but these intermediate steps are not grounded in data.

We have moved the nucleation model to a new figure, Fig. 6, which is only invoked during Discussion. We agree with the reviewer that without identification of the coiled-coil and functional assays we cannot make statements about the order of intermediate steps. We have therefore split the pathway in two: 1. In the top half the coiled-coil first associates with open/closed γ TuRC (ii), which then recruits α/β -tubulin dimers and stabilises longitudinal interactions between γ -tubulin and α/β -tubulin dimers, leading to microtubule nucleation, lattice formation and γ TuRC closure. 2. In the bottom half α/β -tubulin dimers bind transiently to the γ -tubulin ring of open/closed γ TuRC complexes. The coiled-coil proteins stabilise these transient lateral interactions, leading to microtubule nucleation, lattice formation and γ TuRC closure.

In the figure legend we describe a third, less likely scenario where the coiled-coil protein binds free α/β -tubulin first and stabilises lateral interactions with γ -tubulin when that α/β -tubulin dimer interacts with γ TuRC. Given the large excess of α/β -tubulin molecules in the cell, it is unlikely that the coiled-coil-associated α/β tubulin would effectively compete for binding to γ TuRC. We have therefore not included this scenario in the model.

We believe that suggesting a functional model based on new results in the Discussion section of the manuscript would help to stimulate the field and lead to further experiments.

4. Finally, Figure 4d is similarly speculative and not ground in their structural data. The authors should only be showing models that is supported by experimental data.

This Figure panel has been removed.

Reviewer #3:

Remarks to the Author: In this article Dendooven et al report the cryo-ET structure of the γ -Tubulin Ring Complex (γ TuRC) from yeast. γ TuRC were from enriched mitotic spindles. In their preparation, γ TuRC formed a closed conformation and included microtubules (MTs). The overall architecture of γ TuRC resolved here was similar to previously published ones, i.e. seven γ TuSC (made of γ tubulin, Spc97/98) forming a ring. The main difference to previous studies was the diameter of the ring. In addition, the resolved cryo-ET structure has extra densities, one corresponding to unidentified protein that seems to adopt a coiled-coil structure and binds to the first row of α/β -tubulin. The second density, an elongated luminal density that bridges γ TuSC. The authors suggest that this second density is three copies of the N-terminal helical domain of Spc98 (Spc98NHD), based on an AlphaFold trimer prediction and based on their crystal structure of monomeric Spc98NHD. Finally, the authors propose a mechanism for γ TuRC assembly and MTs nucleation (Figure 4c and d). The data presented and image processing is of high quality and I do not have any concern about the presented structure. The weakness of this study is the limited resolution achieved (~8-9Å), which meant the 'coiled-coil' density remained unassigned and the assignment for the luminal density is missing further validation, especially that the resolution of that part of the map looks even worse. Taken that the 'coiled-coil' density is the only major novel aspect of this structure, I am afraid the impact of this structure is very limited. Assuming it is not possible to get higher resolution structure, the authors should do cross-linking mass spectrometry to identify this density.

We thank the reviewer for critically assessing the manuscript, suggesting valuable experiments and ways to improve the clarity and accessibility of our manuscript.

We collaborated with the group of Juri Rappsilber to use cross-linking mass spectrometry (CLMS) of the enriched SPB preps to identify the coiled-coil protein. The Rappsilber's group have pioneered CLMS methods for assemblies both *in vitro* and more recently *in situ* (bacterial cells). Despite the challenge of the complex SPB sample, Rappsilber's group revealed many cross-links that are consistent with the known interactions of γ TuRC, SPB plaques and kinetochores (Fig. 3d). Although CLMS did not detect any other proteins interacting directly with either Spc98 or γ -tubulin, three proteins were identified as having previously unknown cross-links to Spc110, two of which are coiled-coil proteins: Bre1 and Stu2. Only Stu2 is enriched at SPBs, crosslinking near the Spc110 N-terminus, but not directly via its coiled-coil (Fig. 3d and Extended Data Fig. 5a, b). We discuss the implications in the text (page 5, from line 45).

We respectively disagree with the comment regarding the impact of this structure. This is the first relatively high-resolution structure of a native γ TuRC in an active state bound to a nucleated microtubule. The structure explains how γ TuRCs are activated, how they transition from an open low-activity to a closed active state, and shows that γ TuRC is a perfect template for a 13-protofilament microtubule, and that a coiled-coil protein stabilises the first row of α/β -tubulin of a MT attached to γ TuRC.

Minor comments are related to the way the data is presented. The manuscript in its current form assumes previous knowledge on γ TuRC, there are many abbreviations of domains and sub-complexes that are not annotated in the main figures (only in the supplement).

Below are specific suggestions to make it more accessible for the general reader:

1. Spokes and γ TuSC should be annotated clearly in Figure 1.

Thank you for this suggestion. Spokes are numbered. We outlined a γ TuSC protomer with a red box – Fig. 1d (left panel).

2. Incorporate Extended Data Fig. 2a and b in Figure 1.

This is a good suggestion and something that we attempted in earlier versions of the manuscript. However this made Fig. 1 too crowded and in our view detracted from the main

point of Fig. 1, which is the overall structure of the budding yeast γ TuRC. Therefore, we think keeping Extended Data Fig. 2a,b separate will make the manuscript easier to follow.

3. Figure 2b-d can be in the supplement and the font should be bigger for the annotations.

The font size of the labels in Fig. 2 has been increased. One of the main findings of this study, and other recent studies either published (Llorca and Surrey) or in preprint (Kapoor), is the reorganisation of the GRIP2- γ -Tubulin domains upon γ TuRC activation. We therefore would prefer to keep these panels in the main figures.

4. Figure 4d should be removed it is duplication of c(iii)

This Figure panel has now been removed.

5. Extended Data Fig. 6 should be a main figure and the model in b more visible

As suggested, we increased the map transparency in panel b to make the model more visible, and moved both panels a and b into Fig. 5.

6. Line 257, the call for Extended Data Fig. 5b is not correct

We thank the reviewer for spotting this error and have corrected it to Extended Data Fig. 5c.

7. Lastly, Extended Data Tables need consistency when using dots/commas/space with the numbers. Look for example on 'Initial/final particle images (no.)', also is it number of particles or number of 'images' (micrographs)? At

We have revised Extended Tables 1 and 2 to increase consistency and have changed 'Initial/final particle images (no.)' to 'final particles (N)'.

Decision Letter, first revision:

Message: Our ref: NSMB-A48458A

28th Feb 2024

Dear Dr. Barford,

Thank you for submitting your revised manuscript "Structure of the native γ -Tubulin Ring Complex capping spindle microtubules" (NSMB-A48458A). It has now been seen by the original referees and their comments are below. The reviewers find that the paper has improved in revision, and therefore we'll be happy in principle to publish it in Nature Structural & Molecular Biology, pending minor revisions to satisfy the referees' final requests and to comply with our editorial and formatting guidelines.

Sincerely,
Kat

Katarzyna Ciazynska, PhD
(she/her)
Associate Editor
Nature Structural & Molecular Biology
<https://orcid.org/0000-0002-9899-2428>

Reviewer #1 (Remarks to the Author):

The authors have addressed all of my concerns and I am happy for the paper to be published. I take this opportunity to congratulate the authors on a job well done, it's a very nice paper.

Reviewer #2 (Remarks to the Author):

Thank you for your revised manuscript - I have no further comments.

Reviewer #3 (Remarks to the Author):

The authors addressed adequately all my points.

Author Rebuttal, first revision:

There are no additional reviewer requests.

Final Decision Letter:**Message:** 19th Mar 2024

Dear Dr. Barford,

We are now happy to accept your revised paper "Structure of the native γ -Tubulin Ring Complex capping spindle microtubules" for publication as an Article in Nature Structural & Molecular Biology.

Your paper will be published online soon after we receive proof corrections and will appear

in print in the next available issue. You can find out your date of online publication by contacting the production team shortly after sending your proof corrections.

Please note that *Nature Structural & Molecular Biology* is a Transformative Journal (TJ). Authors may publish their research with us through the traditional subscription access route or make their paper immediately open access through payment of an article-processing charge (APC). Authors will not be required to make a final decision about access to their article until it has been accepted. Find out more about Transformative Journals

Authors may need to take specific actions to achieve compliance with funder and institutional open access mandates. If your research is supported by a funder that requires immediate open access (e.g. according to Plan S principles) then you should select the gold OA route, and we will direct you to the compliant route where possible. For authors selecting the subscription publication route, the journal's standard licensing terms will need to be accepted, including self-archiving policies. Those licensing terms will supersede any other terms that the author or any third party may assert apply to any

version of the manuscript.

Sincerely,
Kat

Katarzyna Ciazynska, PhD
(she/her)
Associate Editor
Nature Structural & Molecular Biology
<https://orcid.org/0000-0002-9899-2428>